# Matching without Group Barrier for Heterogeneous Treatment Effect Estimation

**Yuguang Yan**[1]**, Haolin Yang**[1]**, Shihao Zhang**[2]**, Weilin Chen**[1]**, Ruichu Cai**[1,3]***Zhifeng Hao**[4]

[1] School of Computer Science, Guangdong University of Technology, Guangzhou, China,
[2] School of Computing, National University of Singapore, Singapore,
[3] Pazhou Laboratory (Huangpu), Guangzhou, China,
[4] College of Science, Shantou University, Shantou, China.

## Abstract

In heterogeneous treatment effect estimation from observational data, the fundamental challenge is that only the factual outcome under the received treatment is observable, while the potential outcomes under other treatments or no treatment can never be observed. As a simple and effective approach, matching aims to predict counterfactual outcomes of the target treatment by leveraging the nearest neighbors within the target group. However, due to limited observational data and the distribution shifts between groups, one cannot always find sufficiently close neighbors in the target group, resulting in inaccurate counterfactual prediction because of the manifold structure of data. To address this, we remove group barriers and propose a matching method that selects neighbors from all samples, not just the target group. This helps find closer neighbors and improves counterfactual prediction. Specifically, we analyze the effect estimation error in matching, which motivates us to propose a self optimal transport model for matching. Based on this, we employ an outcome propagation mechanism via the transport plan for counterfactual prediction, and exploit factual outcomes to learn a distance as the transport cost. The experiments are conducted on both binary and multiple treatment settings to evaluate our method.

## 1 Introduction

Estimating heterogeneous treatment effects from observational data has been widely applied in many real-world data applications (Hitsch et al., 2024), such as healthcare (Foster et al., 2011), economics (Heckman, 2000), and recommendation systems (Sato et al., 2020; Luo et al., 2024; Gao et al., 2024). Based on the framework of the Neyman-Rubin potential outcome model, the treatment effect can be estimated by comparing the potential outcomes of different treatments (Splawa-Neyman et al., 1990; Rubin, 2005). Nevertheless, we can only observe the factual outcome of the received treatment, while counterfactual outcomes under other treatments or no treatment can never be obtained.

To predict counterfactual outcomes, a variety of machine learning methods have been proposed (Johansson et al., 2016; Feuerriegel et al., 2024). Among them, matching has attracted significant attention because of its simplicity and interpretability (Stuart, 2010; Kallus, 2020). To predict the counterfactual outcome of a target treatment, classical matching identifies the nearest neighbors in the group receiving the target treatment, and then aggregates their factual outcomes for prediction (Kallus, 2020). The cornerstone underlying matching is the assumption that samples close in distance tend to have similar potential outcomes.

However, in practice, due to limited observational data and distribution discrepancies between groups caused by the confounding bias (Greenland et al., 1999; Shalit et al., 2017), there exist regions where samples under the target treatment are scarce or even absent, making it difficult to find sufficiently close samples within the target group. Consequently, the matched samples may suffer from large distances. Since data samples typically lie on an intrinsic manifold, where the Euclidean distance is meaningful only locally, large distances between matched samples may not

---

*Corresponding author: cairuichu@gmail.com

capture true relationships. This inconsistency weakens counterfactual prediction. In other words, matching performs well only when samples are close enough.

To address the above challenge, we propose to remove the barriers between groups and design a matching method to find neighbors from all the samples regardless of their received treatments. By doing this, closer samples with small distances can be matched, which is beneficial for capturing relations between samples for counterfactual prediction. Specifically, we analyze the outcome estimation error of our matching method and provide an error bound in terms of the sample distances. Our theoretical result enjoys an explanation from the perspective of optimal transport, which studies how to move masses from a group of samples to another group with the minimal total transport cost (Villani et al., 2009; Peyré et al., 2019). Motivated by this explanation, we propose a self optimal transport model to select neighbors from all the samples for matching.

Nevertheless, for the matched samples not coming from the target group, their potential outcomes under the target treatment are unknown, bringing a challenge to counterfactual prediction. To alleviate this, inspired by the information propagation mechanism used in semi-supervised learning (Zhu and Ghahramani, 2002), we construct a transition probability matrix based on the optimal transport plan, allowing us to employ a random walk algorithm (Xia et al., 2019) for counterfactual prediction.

To preserve the relations between factual outcomes in the transport cost of our model, we introduce factual outcomes to learn a distance as the transport cost within the optimal transport framework, in which the transport cost measured on covariates is consistent with the optimal transport plan of factual outcomes. To evaluate the performance of our method, we conduct experiments on both semi-synthetic data and simulation datasets, including both binary and multiple treatment settings. We name our method as **M**atching with**O**ut **G**roup b**A**rrier (MOGA), and summarize the major contributions as follows.

- We propose a matching method to select neighbors from all the samples, which is formulated as a self optimal transport model, allowing closer samples to be matched for better capturing sample relationships.

- We propose a counterfactual prediction approach for estimating heterogeneous treatment effects, using an outcome propagation mechanism and the optimal transport plan modeled as a transition probability matrix.

- We propose a distance learning method that improves causal effect estimation by leveraging factual outcomes within the optimal transport framework.

## 2 BACKGROUNDS

In this section, we first present the notations used in the paper, and then provide the background of optimal transport and heterogeneous treatment effect estimation. The comprehensive review of related work on causal effect estimation and optimal transport is provided in the Appendix A.

Given a vector $\mathbf{q} \in \mathbb{R}^N$, $q_i$ is the $i$-th entry. $\mathbf{1}$ represents a vector or matrix with all the entries being 1. The probability simplex $\Sigma_N$ is defined as $\Sigma_N = \{\mathbf{q} \in (\mathbb{R}^+)^N \mid \sum_{i=1}^N q_i = 1\}$. For a matrix $\mathbf{A}$, $\mathbf{A}^\top$ is the transpose of $\mathbf{A}$, and $A_{ij}$ is the $(i,j)$-th entry. For the probability distribution $\mathbf{A} \in (\mathbb{R}^+)^{N \times N}$, the entropy is defined as $H(\mathbf{A}) = -\sum_{i=1}^N \sum_{j=1}^N A_{ij}(\log A_{ij} - 1)$.

### 2.1 OPTIMAL TRANSPORT

Given the sets of probability measures $P(\mathcal{U})$ and $P(\mathcal{V})$ on the spaces $\mathcal{U}$ and $\mathcal{V}$, respectively, and a cost function $c : \mathcal{U} \times \mathcal{V} \to \mathbb{R}^+$. Let $\alpha \in P(\mathcal{U})$ and $\beta \in P(\mathcal{V})$ be two distributions with the samples $u \in \mathcal{U}$ and $v \in \mathcal{V}$. The Kantorovich problem of optimal transport aims to find the optimal probabilistic coupling $\gamma \in P(\mathcal{U} \times \mathcal{V})$ by solving the following problem

$$\min_{\gamma} \int_{\mathcal{U} \times \mathcal{V}} c(u,v)d\gamma(u,v) \quad \text{s.t. } \gamma \in \Gamma(\alpha, \beta), \tag{1}$$

where $\Gamma(\alpha, \beta) \subset P(\mathcal{U} \times \mathcal{V})$ is the set of probabilistic couplings with marginal distributions $\alpha$ and $\beta$.

One of the advantages of optimal transport is that it can be performed without knowing the underlying distribution. Optimal transport can work even on discrete distributions represented by empirical samples. Specifically, for the discrete situation, given the observed samples $\{u_i\}_{i=1}^{n_a}$ and $\{v_i\}_{i=1}^{n_b}$ with $n_a$ and $n_b$ being the numbers of samples, respectively, let $\delta(u_i)$ (*resp.*, $\delta(v_i)$) be the Dirac function at the location $u_i$ (*resp.*, $\delta(v_i)$). The vectors $\boldsymbol{a} \in \Sigma_{n_a}$ and $\mathbf{b} \in \Sigma_{n_b}$ are the probability simplexes, and the $i$-th entry $a_i$ (*resp.*, $b_i$) is the probability masses associated with the sample $u_i$ (*resp.*, $v_i$). Based on the above notations, the empirical distributions can be written as

$$\hat{\alpha} = \sum_{i=1}^{n_a} a_i \delta(u_i), \quad \hat{\beta} = \sum_{i=1}^{n_b} b_i \delta(v_i). \tag{2}$$

Let $\mathbf{C}$ be the cost matrix with the entry $C_{ij} = c(u_i, v_j)$, and $\boldsymbol{\gamma}$ be the transport matrix belonging to the set

$$\Gamma(\hat{\alpha}, \hat{\beta}) = \{\boldsymbol{\gamma} \in (\mathbb{R}^+)^{n_a \times n_b} \mid \boldsymbol{\gamma} \mathbf{1}_{n_b} = \boldsymbol{a}, \boldsymbol{\gamma}^\top \mathbf{1}_{n_a} = \mathbf{b}\}, \tag{3}$$

the discrete form of optimal transport reads

$$\min_{\boldsymbol{\gamma}} \ \langle \mathbf{C}, \boldsymbol{\gamma} \rangle \quad \text{s.t.} \ \boldsymbol{\gamma} \in \Gamma(\hat{\alpha}, \hat{\beta}). \tag{4}$$

## 2.2 Causal Effect Estimation

Our analysis follows the Neyman-Rubin potential outcomes framework (Rubin, 1974; Splawa-Neyman et al., 1990). We denote $t_i$ as the treatment received by the $i$-th sample, and $t$ as a treatment value in the space $\mathcal{T} = \{0, 1, \ldots, T\}$, where $T$ is the number of the different treatment values, and $0$ indicates the control group received no treatment. The samples are represented as $\{(\mathbf{x}_i, y_i, t_i)\}_{i=1}^n$, where $n$ is the number of samples, $\mathbf{x}_i \in \mathbb{R}^d$ is the covariate vector with $d$ being the number of covariates, $y_i \in \mathbb{R}$ is the observed factual outcome and $t_i \in \mathcal{T}$ is the received treatment. For the treatment group $t$, the samples are represented as $\{(\mathbf{x}_i^t, y_i^t)\}_{i=1}^{n_t}$ with $n_t$ being the number of samples in the treatment group $t$. Further, we denote $Y_t(\mathbf{x}_i)$ as the potential outcome for the specific individual $i$ given its covariates under the treatment $t$.

Our task is to estimate the heterogeneous treatment effect (HTE), which captures how the impact of a treatment differs based on individual characteristics. In this paper, we focus on the multiple treatment setting, thus the task is to estimate all HTEs under all possible treatments. Formally, for a given treatment $t$ and $t'$, HTE is defined as:

$$\tau_{t,t';i} = \mathbb{E}[Y_t(\mathbf{x}_i) - Y_{t'}(\mathbf{x}_i)|\mathbf{x}_i] \tag{5}$$
$$= f_t(\mathbf{x}_i) - f_{t'}(\mathbf{x}_i), \tag{6}$$

where we denote nuisance function under treatment $t$ as $f_t(\mathbf{x}_i) := \mathbb{E}[Y_t(\mathbf{x}_i)|\mathbf{x}_i]$. Following (Yan et al.; Scotina and Gutman, 2019; Schwab et al., 2018), we make the following assumptions to ensure the identification:

**Assumption 1** (Stable Unit Treatment Value Assumption). The potential outcome of a unit is unaffected by the treatment status of other units, and there is no variation in the treatment levels.

**Assumption 2** (Unconfoundedness). For the $i$-the sample, the received treatment $t_i$ is independent of the potential outcomes $Y_t(\mathbf{x})$ conditioned on the covariates $\mathbf{x}_i$. Formally, $\forall t \in \{0, 1, \ldots, T\}$, $Y_t(\mathbf{x}_i) \perp\!\!\!\perp t_i | \mathbf{x}_i$.

**Assumption 3** (Overlap). For each sample, there is a non-zero probability of being assigned either treatment or control, conditional on the covariates $\mathbf{x}_i$. Formally, $\forall t \in \{0, 1, \ldots, T\}$ and $\mathbf{x}_i$, we have $0 < P(t \mid \mathbf{x}_i) < 1$.

**Assumption 4** (Lipschitz Continuity). Given the mapping function $\phi : \mathbb{R}^d \to \mathbb{R}^{d'}$ and the norm $\|\mathbf{x}_i - \mathbf{x}_j\|_\phi = \|\phi(\mathbf{x}_i) - \phi(\mathbf{x}_j)\|$. For any treatment $t \in \mathcal{T}$, the nuisance function $f_t(\cdot)$ is Lipschitz continuous with the constant $L_t > 0$, *i.e.*, $|f_t(\mathbf{x}_i) - f_t(\mathbf{x}_j)| \leq L_t \|\mathbf{x}_i - \mathbf{x}_j\|_\phi$.

Assumptions 1, 2, and 3 are common and standard assumptions in causal inference (Rubin, 1974; Hill, 2011; Johansson et al., 2016), which guarantee the identification of HTE. Here we adopt the classic assumptions under the standard identification framework. Additionally, we make an assumption in Assumption 4 on the function $f_t(\mathbf{x})$, which ensures that the function $f_t$ does not change too rapidly and provides a bound on how much the potential outcome varies as $\mathbf{x}$ changes. Intuitively, Assumption 4 means that two close samples usually have similar potential outcomes. This assumption is reasonable and easy to be satisfied in practice (Kallus, 2020).

## 3 METHODOLOGY

### 3.1 MATCHING WITHOUT GROUP BARRIER

Given a sample $\mathbf{x}^{t'}$ with $t' \neq t$, in order to predict its counterfactual outcome under the treatment $t$, classical matching finds the nearest neighbors from the treatment group $t$ $\{(\mathbf{x}_i^t, y_i^t)\}_{i=1}^{n_t}$ based on a distance $d(\mathbf{x}^{t'}, \mathbf{x}_i^t)$, and then combines the factual outcomes of the neighbors to predict the counterfactual outcome $\hat{y}^t$. The assumption underlying matching is that if $d(\mathbf{x}^{t'}, \mathbf{x}_i^t)$ is small, then they have similar potential outcomes. However, this approach faces a challenge in practice that one cannot always find sufficiently close neighbors in the treatment group $t$. This occurs when the number of observational samples is limited, or the groups $t$ and $t'$ suffer from a large distribution discrepancy because of the confounding bias. Although one can find a neighbor with a large distance, the manifold structure of data makes large distances unreliable to accurately characterize the structure of potential outcomes, resulting in inaccurate counterfactual prediction.

To address this, we break the barriers between different groups and propose a matching method to find neighbors from all the samples regardless of their received treatments, so that smaller distances between matched samples are expected, improving the reliability of counterfactual prediction. Specifically, to predict the counterfactual outcome of the treatment $t$ for the sample $\mathbf{x}_i$, we find nearest neighbors of $\mathbf{x}_i$ from all the samples rather than the treatment group $t$, and then combine the potential outcomes of the matched neighbors for prediction. Without loss of generality, let $W_{ij}$ be the matching degree between $\mathbf{x}_i$ and $\mathbf{x}_j$, the counterfactual outcome can be estimated by the following

$$\hat{Y}_t(\mathbf{x}_i) = \sum_{j=1}^{n} W_{ij} Y_t(\mathbf{x}_j), \tag{7}$$

where $\sum_{j=1}^{n} W_{ij} = 1$. Ideally, $W_{ij}$ of the matched neighbors should be large, and $W_{ij}$ of the non-matched samples should be small or close to $0$. In addition, $W_{ii}$ is expected to be zero since the outcome estimation of $x_i$ relies on its neighbors rather than itself. The consistency analysis of the estimator is discussed in Appendix B. The outcome estimation error of the treatment $t$ is analyzed by the following theorem with an upper error bound:

**Theorem 1.** *Let $\epsilon_{Y_t} = \sum_{i=1}^{n} (f_t(\mathbf{x}_i) - \hat{Y}_t(\mathbf{x}_i))^2$ be the estimation error of the potential outcomes under the treatment $t$, if the assumption $\forall i, Var(y_i|\mathbf{x}_i, t_i) = \eta^2$ holds, then the error is upper bounded by the following:*

$$\epsilon_{Y_t} \leq 2L_t^2 \sum_{i=1}^{n} \sum_{j=1}^{n} W_{ij} \|\phi(\mathbf{x}_i) - \phi(\mathbf{x}_j)\|_2^2 + 2n\eta^2 \sum_{i=1}^{n} \sum_{j=1}^{n} W_{ij}^2. \tag{8}$$

The proof is in Appendix C. The homogeneity assumption of the variance is used in (Kuang et al., 2019; Kallus, 2020).

Based on Theorem 1 we discuss the difference between our matching method and the classical matching method from the perspective of outcome estimation error. To predict counterfactual outcome $Y_t(\mathbf{x}_i)$, no matter which treatment $t$ is considered, our matching method is able to find neighbors from all the samples $\{\mathbf{x}_j\}_{j=1}^{n}$ regardless of their received treatment $\{t_j\}_{j=1}^{n}$. On the contrary, classical matching only considers the target group $\{\mathbf{x}_j : t_j = t\}$ as the candidates, which highly restricts the search space for matching. As a result, the neighbors in the other group are excluded, which suffers from larger distances between $\mathbf{x}_i$ and the matched samples, resulting in a looser upper bound.

The upper bound in Theorem 1 enjoys a clear explanation from the perspective of optimal transport. The first term can be modeled as the total transport cost, and the second term can be modeled as the Frobenius norm of the transport matrix, motivating us to propose a regularized optimal transport model for learning the matching degree matrix. Specifically, we define the empirical distribution of all the samples as $\mu = \sum_{i=1}^{n} p_i \delta(\mathbf{x}_i), p_i = \frac{1}{n}, \forall i = 1, \ldots, n$, where the uniform probability mass $p_i$ indicates that all the samples contribute equally to the estimation error. The uniform masses also prevent large subgroups from dominating small subgroups, which ensures that internal subgroups maintain influence in the matching process. By setting the probabilistic coupling as $\gamma_{ij} = \frac{1}{n} W_{ij}$, we

minimize the upper bound of the potential outcome estimation error in Theorem 1 by the following optimal transport problem

$$\min_{\boldsymbol{\gamma}} \langle \mathbf{C}^\phi, \boldsymbol{\gamma} \rangle + \lambda_f \Omega(\boldsymbol{\gamma}) \qquad \text{s.t. } \boldsymbol{\gamma} \in \Gamma(\mu, \mu), \ \gamma_{ii} = 0, \forall i = 1, \ldots, n, \qquad (9)$$

where $\mathbf{C}^\phi$ is the cost matrix with the $(i, j)$-th entry being $C_{ij}^\phi = \|\phi(\mathbf{x}_i) - \phi(\mathbf{x}_j)\|_2^2$, $\Omega(\boldsymbol{\gamma}) = \frac{1}{2}\|\boldsymbol{\gamma}\|_F^2$ is the square of the Frobenius norm, and $\lambda_f$ is the trade-off hyperparameter.

Different from the classical optimal transport model involving two distributions discussed in Section 2.1, Problem (9) is a self optimal transport model that considers transport from the set of samples to this set while excluding moving one sample to itself (Landa et al., 2021; Yan et al., 2024). As a result, for the sample $\mathbf{x}_i$, the neighbors found from all the samples are matched with a large weight $W_{ij}$, while the samples far away from $\mathbf{x}_i$ are assigned with a weight $W_{ij}$ close to 0.

The following theorem further shows that by minimizing the upper bound in Theorem 1, the effect estimation error is also minimized

**Theorem 2.** *Let $\hat{Y}_t(\mathbf{x}_i)$ denote the predicted outcome for the $i$-th sample under the treatment $t$. The effect estimation error is measured by the pairwise precision in the estimation of heterogeneous effect (mPEHE) defined as $\epsilon_{mPEHE} = \frac{2}{nT(T+1)} \sum_{0 \leq t' < t \leq T} \sum_{i=1}^n ((\hat{Y}_t(\mathbf{x}_i) - \hat{Y}_{t'}(\mathbf{x}_i)) - (f_t(\mathbf{x}_i) - f_{t'}(\mathbf{x}_i)))^2$ (Schwab et al., 2018; Guo et al., 2023). The effect estimation error is upper bounded by the outcome estimation error as follows*

$$\epsilon_{mPEHE} \leq \frac{4}{n(T+1)} \sum_{t=0}^T \epsilon_{Y_t}. \qquad (10)$$

The proof is given in Appendix D. We observe that the bound of in Theorem 2 is upper bounded in Theorem 1.

Consequently, we establish a theoretical connection between our matching method and optimal transport. In practice, to remove the constraints $\gamma_{ii} = 0$, we follow (Yan et al., 2024) to construct a cost matrix $\tilde{\mathbf{C}}^\phi = \mathbf{C}^\phi + L\mathbf{I}_n$, where $L$ is a sufficiently large value and $\mathbf{I}_n \in \mathbb{R}^{n \times n}$ is the identity matrix. By doing this, the diagonal entries $\tilde{\mathbf{C}}^\phi$ will induce $\gamma_{ii}$ to be close to 0, avoiding tackling the constraints $\gamma_{ii} = 0$ explicitly. In addition, we borrow the entropic regularization term $H(\boldsymbol{\gamma})$ to minimize the negative entropy of $\boldsymbol{\gamma}$, so that the Sinkhorn algorithm (Cuturi, 2013) can be applied to efficiently solve the optimal transport problem. Finally, we achieve the following optimal transport problem:

$$\min_{\boldsymbol{\gamma}} \ \langle \tilde{\mathbf{C}}^\phi, \boldsymbol{\gamma} \rangle + \lambda_f \Omega(\boldsymbol{\gamma}) - \lambda_h H(\boldsymbol{\gamma}) \qquad \text{s.t. } \boldsymbol{\gamma} \in \Gamma(\mu, \mu), \qquad (11)$$

where $\lambda_h$ is the hyperparameter.

Based on our optimal transport model, we leverage the results of our model for counterfactual prediction in Section 3.2, and incorporate factual outcomes to learn a distance as the transport cost in Section 3.3.

## 3.2 COUNTERFACTUAL PREDICTION

Given the optimal transport plan $\boldsymbol{\gamma}$ obtained by solving Problem (11), we can find the matched samples and predict the counterfactual outcome $Y_t(\cdot)$ according to Eq. (7). The optimal transport plan reflects the matching degrees used for counterfactual outcome estimation. Optimal transport will adaptively assign larger values $\gamma_{ij}$ between close sample pairs, and a pair far away from each other will receive a quite small $\gamma_{ij}$. For outliers that are far away from most samples, optimal transport will assign small weights to them. As a result, the estimated outcomes are basically determined by close samples with large weights, and the outliers will make a limited contribution in counterfactual outcome estimation. However, for the matched samples not come from the treatment group $t$, their potential outcomes under the treatment $t$ are unknown. To tackle this, we consider an information propagation mechanism to iteratively update the counterfactual predictions by a random walk method (Xia et al., 2019).

Recall that $\gamma \in \Gamma(\mu, \mu)$ is a doubly stochastic matrix, meaning that $\sum_{j=1}^{n} \gamma_{ij} = \frac{1}{n}, \forall i = 1, \ldots, n$. We can simply construct a transition probability matrix $\mathbf{W} \in (\mathbb{R}^{+})^{n \times n}$ by setting $\mathbf{W} = n\gamma$. The $(i, j)$-th entry $W_{ij}$ indicates the probability that the $i$-th sample moves to the $j$-th sample, where the probability is measured based on the transport cost between them compared with the costs between other pairs. Based on this, we develop a random walk algorithm to predict potential outcomes for all the treatments over all the samples.

Specifically, let $\mathbf{Y} \in \mathbb{R}^{n \times (T+1)}$ be the matrix including all the factual outcomes of all the samples, which is defined as

$$Y_{it} = \begin{cases} y_i & \text{for} \quad t_i = t. \\ 0 & \text{for} \quad t_i \neq t, \end{cases} \tag{12}$$

and $\mathbf{M} \in \{0, 1\}^{n \times (T+1)}$ be the factual outcome mask matrix defined as

$$M_{it} = \begin{cases} 1 & \text{for} \quad t_i = t, \\ 0 & \text{for} \quad t_i \neq t. \end{cases} \tag{13}$$

At the $\kappa$-th iteration, we use $\widehat{\mathbf{Y}}^{\kappa} \in \mathbb{R}^{n \times (T+1)}$ to denote the predicted potential outcome matrix including all the treatments and samples. We update the predicted potential outcome matrix by $\mathbf{S}\widehat{\mathbf{Y}}^{\kappa}$ where $\mathbf{S}$ is the affinity matrix constructed as $\mathbf{S} = \rho\mathbf{W} + (1 - \rho)\mathbf{I}$, which introduces self-connections with the coefficient $\rho \in (0, 1)$, which balances exploration (via $\mathbf{W}$) and memory (via $\mathbf{I}$). After that, we replace the predicted entries $\hat{Y}_{it}^{\kappa}$ with the known corresponding factual outcome by $Y_{it}$. In summary, we iteratively update the predicted potential outcome matrix by the following random walk rule

$$\widehat{\mathbf{Y}}^{\kappa+1} = \mathbf{S}\widehat{\mathbf{Y}}^{\kappa} \odot (\mathbf{1} - \mathbf{M}) + \mathbf{Y} \odot \mathbf{M} = \mathbf{S}\widehat{\mathbf{Y}}^{\kappa} \odot (\mathbf{1} - \mathbf{M}) + \mathbf{Y}, \tag{14}$$

and the initial potential outcome matrix is set as $\widehat{\mathbf{Y}}^0 = \mathbf{Y}$.

Eq. (14) contains a diffusion term $\mathbf{S}\widehat{\mathbf{Y}}^{\kappa} \odot (\mathbf{1} - \mathbf{M})$, which gradually propagates outcome information along the manifold to the unmasked part denoted by $\mathbf{1} - \mathbf{M}$, mimicking geodesic aggregation and ensuring a smooth geodesic estimation (Tenenbaum et al., 2000). By doing this, the manifold structure of data is leveraged to improve the causal effect estimation. Finally, the outcome information will gradually propagate to all the samples under all the treatments. The fixed observation term $\mathbf{Y} \odot \mathbf{M}$ remains unchanged across iterations.

### 3.3 DISTANCE LEARNING

Based on Theorem 1, the estimation error of heterogeneous treatment effects relies on the cost $c_\phi(\mathbf{x}_i, \mathbf{x}_j) = \|\phi(\mathbf{x}_i) - \phi(\mathbf{x}_j)\|_2^2$ which is determined by $\phi(\cdot)$. In this part, we discuss how to implement the function $\phi(\cdot)$.

The vanilla approach is the identity function $\phi(\mathbf{x}) = \mathbf{x}$, and the cost $c_{id}(\mathbf{x}_i, \mathbf{x}_j) = \|\mathbf{x}_i - \mathbf{x}_j\|_2^2$ is the squared Euclidean distance, which is commonly used in existing works of optimal transport (Courty et al., 2017).

The key to the success of matching is to find a sample with similar potential outcomes, which means that $c_\phi(\mathbf{x}_i, \mathbf{x}_j)$ can capture the difference between their potential outcomes. However, $c_{id}(\cdot, \cdot)$ does not take the outcome into consideration. To enhance the distance measurement for potential outcome prediction, we introduce the factual outcomes into distance learning. In addition, since the distance is adopted as the transport cost in our optimal transport model in Problem (9), we also apply the framework of optimal transport to learn a distance. Specifically, we consider sample transport within each treatment group, which involves the transport plans $\{\gamma_t\}_{t=0}^{T}$, with the constraint that $\gamma_t \in \Gamma(\mu_t, \mu_t)$, where the empirical distribution of one group $\mu_t$ is defined as $\mu_t = \sum_{i=1}^{n_t} p_i^t \delta(\mathbf{x}_i^t), p_i^t = \frac{1}{n_t}, \forall i = 1, \ldots, n_t$.

Based on the above discussions, we first exploit factual outcomes to learn an optimal transport plan for each group, and then enforce the learned distance on covariates to admit the same optimal

transport plans obtained from factual outcomes. Specifically, we learn the optimal transport plan based on factual outcomes by the following problem

$$\tilde{\boldsymbol{\gamma}}_t = \arg\min_{\boldsymbol{\gamma}_t} \ \langle \mathbf{C}_t^Y, \boldsymbol{\gamma}_t \rangle - \lambda_h H(\boldsymbol{\gamma}_t) \qquad \text{s.t. } \boldsymbol{\gamma}_t \in \Gamma(\mu_t, \mu_t), \tag{15}$$

where the cost matrix $\mathbf{C}_t^Y$ measured by the factual outcomes of the treatment group $t$ is constructed as $C_{t,ij}^Y = (y_i^t - y_j^t)^2$. Similar to the self optimal transport model in (Yan et al., 2024), we set $C_{t,ii}^Y = L$ as a sufficiently large value to avoid the trivial solution. After that, we learn the mapping function $\phi(\cdot)$ based on the optimal transport plan $\tilde{\boldsymbol{\gamma}}_t$ by the following

$$\min_{\phi} \ \sum_{t=0}^{T} \langle \mathbf{C}_t^\phi, \tilde{\boldsymbol{\gamma}}_t \rangle, \tag{16}$$

where $\mathbf{C}_t^\phi$ is the cost matrix determined by the function $\phi(\cdot)$ on the covariates of the treatment group $t$. Intuitively, for the pair of $i$-th and $j$-th samples, if their potential outcomes are similar, a large mass transport $\tilde{\gamma}_{t,ij}$ will be induced, resulting in a small cost $C_{t,ij}^\phi$. Eq. (16) encourages $\mathbf{C}_t^\phi$ to approach $\mathbf{C}_t^Y$, which improves the consistency between $c_\phi(\mathbf{x}_i, \mathbf{x}_j)$ and $\mathbf{C}_{t,ij}^Y$. As a result, the outcome information is effectively captured in the learned cost $c_\phi(\cdot, \cdot)$. Moreover, the ordinal relation of the potential outcomes $\{y_i^t\}_{i=1}^{n_t}$ is well preserved in $\phi(\mathbf{x})$, which has been shown to compress the manifold on which $\phi(\mathbf{x})$ lie, leading to improved generalization ability (Zhang et al., 2024).

To further introduce $\phi(\cdot)$ and $\{\boldsymbol{\gamma}_t\}_{t=0}^T$ into a unified problem, we propose the following self optimal transport model, which considers the transport cost on both covariates and factual outcomes collaboratively, and learn the transport cost and plans jointly

$$\min_{\{\boldsymbol{\gamma}_t\}, \phi} \ \sum_{t=0}^{T} \langle \mathbf{C}_t^\phi, \boldsymbol{\gamma}_t \rangle + \lambda_y \langle \mathbf{C}_t^Y, \boldsymbol{\gamma}_t \rangle - \lambda_h H(\boldsymbol{\gamma}_t) \qquad \text{s.t. } \boldsymbol{\gamma}_t \in \Gamma(\mu_t, \mu_t), \quad t = 0, \ldots, T, \tag{17}$$

where $\lambda_y$ is the trade-off hyperparameter. We initialize the optimal transport plans $\boldsymbol{\gamma}_t$ based on the solution to Problem (15), and then solve Problem (17) to refine $\boldsymbol{\gamma}_t$ shared by the cost of covariates and factual outcomes. As a result, the coupled transport cost $\mathbf{C}_t^\phi$ measured on covariates can be supervised by the factual outcomes.

Now we discuss how to solve Problem (17) to obtain the optimal transport plans $\{\boldsymbol{\gamma}_t\}_{t=0}^T$ and the mapping function $\phi(\cdot)$ involved in the transport cost. Problem (17) contains multiple blocks of parameters. We adopt the alternate method to solve the problem, during which we optimize one block of parameters with the other blocks fixed.

Specifically, given the fixed mapping function $\phi(\cdot)$ and the corresponding cost matrix $\mathbf{C}_t^\phi$, and the cost matrix $\mathbf{C}_t^Y$, the optimal transport problem within each group can be separated and solved individually. For the treatment group $t$, the subproblem with respect to $\boldsymbol{\gamma}_t$ can be formulated as follows

$$\min_{\boldsymbol{\gamma}_t} \ \langle \mathbf{C}_t^\phi, \boldsymbol{\gamma}_t \rangle + \lambda_y \langle \mathbf{C}_t^Y, \boldsymbol{\gamma}_t \rangle - \lambda_h H(\boldsymbol{\gamma}_t) \qquad \text{s.t. } \boldsymbol{\gamma}_t \in \Gamma(\mu_t, \mu_t), \tag{18}$$

which is a standard self optimal transport problem with the cost matrix $\mathbf{C}_t^\phi + \lambda_y \mathbf{C}_t^Y$ and can be solved by the Sinkhorn algorithm (Cuturi, 2013).

Given the fixed transport plans $\{\boldsymbol{\gamma}_t\}_{t=0}^T$, we optimize the mapping function $\phi(\cdot)$ to obtain the coupled cost function. Here, we implement $\phi(\cdot)$ as a projection operation $\phi(\mathbf{x}) = \mathbf{P}^\top \mathbf{x}$ with $\mathbf{P} \in \mathbb{R}^{d \times d'}$ being the parameters to be optimized, so that the transport cost $C_{t,ij}^\phi$ can be obtained as

$$C_{t,ij}^\phi = c_\phi(\mathbf{x}_i, \mathbf{x}_j) = \|\mathbf{P}^\top \mathbf{x}_i - \mathbf{P}^\top \mathbf{x}_j\|_2^2. \tag{19}$$

As a result, the transport cost is directly guided by the factual outcomes, ensuring that the learned distance reflects meaningful relations of samples in the sense that close samples with small learned distance have similar outcomes. In addition, the transport cost is calculated in the supervised subspace, which can alleviate the issue of high-dimensional data.

To avoid the trivial solution and induce orthogonal projected features, we make $\mathbf{P}$ to follow the constraint $\mathbf{P} \in \mathcal{M} = \{\mathbf{P} \in \mathbb{R}^{d \times d'} \mid \mathbf{P}^\top \mathbf{P} = \mathbf{I}\}$. Based on this, the subproblem with respect to $\mathbf{P}$ is given as follows

$$\min_{\mathbf{P}} \ \sum_{t=0}^{T} \langle \mathbf{C}_t^{\mathbf{P}}, \boldsymbol{\gamma}_t \rangle \qquad \text{s.t. } \mathbf{P} \in \mathcal{M}. \tag{20}$$

The following proposition provides the closed-form solution to this problem.

**Proposition 3.** *Let $\mathbf{X}_t \in \mathbb{R}^{n_t \times d}$ be the matrix including all the samples in the treatment group $t$. Problem 20 is equivalent to the following problem*

$$\min_{\mathbf{P}} \ \mathrm{tr}\left(\mathbf{P}^\top (\sum_{t=0}^{T} \boldsymbol{\Theta}_t)\mathbf{P}\right) \quad s.t. \ \mathbf{P}^\top \mathbf{P} = \mathbf{I}, \tag{21}$$

*where the matrix $\boldsymbol{\Theta}_t$ is constructed as*

$$\boldsymbol{\Theta}_t = 2(\mathbf{X}_t)^\top diag(\boldsymbol{\gamma}_t \mathbf{1} - \boldsymbol{\gamma}_t)\mathbf{X}_t. \tag{22}$$

*The closed-form solution to this problem is obtained by the eigenvectors associated with the $d'$ smallest eigenvalues of the matrix $\sum_{t=0}^{T} \boldsymbol{\Theta}_t$.*

The proof is given in Appendix F.

The pseudo-code of our algorithm is given in Appendix G.

## 4 EXPERIMENTS

In this section, we first describe the experimental settings including the compared methods and evaluation metrics. After that, we present experimental results and discussion on semi-synthetic and simulation datasets. More experiments can be found in the appendix, including matching visualization results, ablation studies, and sensitivity analysis. All the experiments can be run on a single 24GB GPU of NVIDIA GeForce RTX 4090.

### 4.1 EXPERIMENTAL SETTINGS

**Compared Methods** We compare the performance of MOGA with the following methods: **k-NN**(Crump et al., 2008) finds $k$ nearest neighbors from the target group and then predicts the potential outcome based on the factual outcomes of the neighbors. **OLS/LR-2** applies linear regression with separate regression models for each treatment group. **BART** (Chipman et al., 2010; Hill, 2011) provides a posterior distribution of the treatment effects, allowing for uncertainty quantification in causal inference tasks. **TARNet** (Shalit et al., 2017) learns latent representations of covariates to reduce the distribution discrepancy between the treated and control groups. **CFR** (Shalit et al., 2017) minimizes the distribution discrepancy between treated and control groups in the latent representation space via the Integral Probability Metric, which is implemented by the Wasserstein distance. We defined regularized all treatments to have the same activation distribution in the topmost shared layer, extending CFR to multiple treatment settings. **GANITE** (Yoon et al., 2018) estimates individual treatment effects using a generative model based on Generative Adversarial Networks. **PSM** (Rosenbaum and Rubin, 1983) estimates the treatment effect by matching individuals in the treated and control groups based on the propensity score, which is predicted by logistic regression. **PM** (Schwab et al., 2018) enhances the matching method by learning a neural network to estimate propensity scores within mini-batches. **CP** (Harada and Kashima, 2021) constructs a graph based on similarities between samples, and then applies a graph-based semi-supervised learning method for causal inference. **GOM** (Kallus, 2020) unifies and extends matching, covariate balancing, and doubly-robust estimation by minimizing a bias–variance trade-off under a general function norm. **KOM** (Kallus, 2020) instantiates GOM with an RKHS norm to achieve robust causal estimates. **CEM** (Iacus et al., 2012) (Coarsened Exact Matching) enhances causal inference by strategically reducing the precision of covariates through data coarsening, followed by exact matching. **MitNet** (Guo et al., 2023) proposes to use mutual information to characterize confounding bias in heterogeneous treatment effect estimation.

Table 1: Result on Semi-synthetic data in terms of mean and standard deviation. A lower metric indicates better performance. We highlight the best results in bold and underline the second-best results.

| Dataset | News-2 | | | News-4 | | | News-8 | | | TCGA | | |
|---|---|---|---|---|---|---|---|---|---|---|---|---|
| metric | $\sqrt{\epsilon_{PEHE}}$ | $\epsilon_{ATE}$ | $\sqrt{AMSE}$ | $\sqrt{\epsilon_{mPEHE}}$ | $\epsilon_{mATE}$ | $\sqrt{AMSE}$ | $\sqrt{\epsilon_{mPEHE}}$ | $\epsilon_{mATE}$ | $\sqrt{AMSE}$ | $\sqrt{\epsilon_{mPEHE}}$ | $\epsilon_{mATE}$ | $\sqrt{AMSE}$ |
| k-NN | 9.418± 2.446 | 1.546± 1.472 | 6.972± 1.861 | 10.081± 1.973 | 2.638± 1.308 | 8.471± 1.824 | 11.469± 1.349 | 3.976± 0.976 | 11.124± 1.698 | 11.410± 0.082 | 3.049± 0.130 | 8.351± 0.080 |
| OLS/LR-2 | 7.751± 0.040 | 1.531± 1.479 | 6.961± 1.850 | 9.720± 2.194 | 3.233± 2.004 | 8.658± 2.169 | 10.454± 1.408 | 3.906± 1.166 | 11.974± 2.136 | 13.669± 0.034 | 8.183± 0.029 | 11.007± 0.038 |
| BART | 9.214± 2.371 | 1.528± 1.479 | 6.898± 1.840 | 9.341± 1.888 | 2.617± 1.276 | 8.058± 1.773 | 11.365± 1.783 | 5.172± 1.597 | 10.853± 2.177 | 10.628± 0.031 | 2.812± 0.082 | 7.839± 0.0396 |
| TARNet | 9.283± 2.357 | 1.614± 1.383 | 6.950± 1.863 | 9.956± 2.375 | 3.383± 1.772 | 8.189± 1.927 | 14.520± 2.287 | 9.375± 1.892 | 9.983± 1.533 | 13.595± 0.113 | 6.951± 0.071 | 11.251± 0.102 |
| CFR | 9.291± 2.376 | 1.578± 1.446 | 6.961± 1.865 | 9.746± 2.186 | 3.386± 1.770 | 8.194± 1.928 | 14.517± 2.293 | 9.372± 1.896 | 9.980± 1.536 | 13.358± 0.054 | 6.917± 0.057 | 11.147± 0.087 |
| GANITE | 10.019± 2.651 | 3.190± 3.119 | 9.074± 2.692 | 9.907± 2.305 | 3.570± 1.997 | 8.633± 2.132 | 10.428± 1.405 | 3.842± 1.048 | 9.195± 1.384 | 13.792± 0.039 | 8.147± 0.068 | 13.266± 0.042 |
| PSM | 14.957± 3.579 | 4.020± 3.263 | 10.736± 2.595 | 15.371± 2.970 | 3.628± 1.919 | 12.331± 2.808 | 17.175± 2.143 | 3.844± 1.011 | 15.616± 2.441 | 16.055± 1.451 | 7.416± 1.728 | 11.780± 1.021 |
| GOM | 6.451± 2.155 | 1.532± 1.479 | 4.561± 1.524 | 7.739± 1.792 | 2.618± 1.2697 | 7.028± 1.692 | 12.857± 2.065 | 4.582± 1.569 | 11.671± 1.755 | 10.545± 0.032 | 2.708± 0.069 | 7.687± 0.037 |
| KOM | 6.451± 2.155 | 1.532± 1.479 | 4.562± 1.524 | 7.740± 1.792 | 2.618± 1.269 | 7.028± 1.692 | 12.074± 1.389 | 3.951± 0.969 | 11.429± 1.714 | 10.836± 0.045 | 2.763± 0.095 | 7.894± 0.044 |
| CEM | 9.472± 2.444 | 1.5071± 1.4589 | 6.961± 1.842 | 10.412± 2.021 | 2.626± 1.294 | 8.664± 1.854 | 12.857± 2.065 | 4.582± 1.569 | 11.669± 1.754 | 11.326± 0.048 | 2.768± 0.096 | 8.2603± 0.0463 |
| PM | 9.340± 2.376 | 1.631± 1.468 | 6.971± 1.854 | 10.098± 2.696 | 3.736± 2.713 | 8.968± 2.638 | 10.514± 1.332 | 3.915± 0.929 | 10.590± 1.688 | 13.472± 0.266 | 7.032± 0.206 | 11.095± 0.365 |
| CP | 10.310 ± 2.716 | 4.005 ± 3.200 | 7.734 ± 2.282 | 9.910± 2.291 | 3.598± 1.942 | 7.315± 1.873 | 13.889± 2.745 | 8.610± 2.427 | 10.134± 1.918 | 11.380± 0.123 | 3.511± 0.211 | 9.204± 0.092 |
| MitNet | 7.382± 2.580 | 2.923± 2.374 | 5.220± 1.824 | 8.003± 2.260 | 2.950± 1.811 | 7.986± 2.632 | 9.282± 1.385 | 3.494± 0.929 | 11.744± 2.194 | 10.715± 0.086 | 3.628± 0.182 | 9.315± 0.058 |
| MOGA | 5.081± 1.693 | 0.449± 0.3437 | 3.591± 1.197 | 5.960± 1.180 | 1.155± 0.706 | 4.420± 0.935 | 8.904± 1.214 | 2.386± 0.725 | 7.819± 1.212 | 10.597± 0.037 | 2.785± 0.075 | 7.751± 0.041 |

Table 2: Result on synthetic data in terms of mean and standard deviation. A lower metric indicates better performance. We highlight the best results in bold and underline the second-best results.

| Dataset | $m = [0.1, 0.2, 0.3, 0.4, 0.5]$ | | | $m = [0.1, 0.3, 0.5, 0.7, 0.9]$ | | | $m = [0.1, 0.4, 0.7, 1.0, 1.3]$ | | | $m = [0.1, 0.5, 0.9, 1.3, 1.7]$ | | |
|---|---|---|---|---|---|---|---|---|---|---|---|---|
| metric | $\sqrt{\epsilon_{PEHE}}$ | $\epsilon_{ATE}$ | $\sqrt{AMSE}$ | $\sqrt{\epsilon_{mPEHE}}$ | $\epsilon_{mATE}$ | $\sqrt{AMSE}$ | $\sqrt{\epsilon_{mPEHE}}$ | $\epsilon_{mATE}$ | $\sqrt{AMSE}$ | $\sqrt{\epsilon_{mPEHE}}$ | $\epsilon_{mATE}$ | $\sqrt{AMSE}$ |
| k-NN | 1.592 ± 0.062 | 0.204 ± 0.088 | 1.222 ± 0.037 | 1.627 ± 0.060 | 0.253 ± 0.069 | 1.238 ± 0.035 | 1.663 ± 0.056 | 0.328 ± 0.107 | 1.259 ± 0.034 | 1.653 ± 0.071 | 0.327 ± 0.117 | 1.258 ± 0.051 |
| OLS/LR-2 | 1.423 ± 0.036 | 0.218 ± 0.134 | 1.323 ± 0.034 | 1.458 ± 0.062 | 0.233 ± 0.156 | 1.338 ± 0.048 | 1.510 ± 0.104 | 0.254 ± 0.266 | 1.361 ± 0.092 | 1.508 ± 0.053 | 0.235 ± 0.191 | 1.349 ± 0.052 |
| BART | 1.432 ± 0.046 | 0.191 ± 0.092 | 1.170 ± 0.030 | 1.486 ± 0.048 | 0.232 ± 0.088 | 1.199 ± 0.030 | 1.510 ± 0.104 | 0.254 ± 0.266 | 1.361 ± 0.092 | 1.540 ± 0.066 | 0.324 ± 0.142 | 1.211 ± 0.042 |
| TARNet | 1.606 ± 0.079 | 0.134 ± 0.077 | 1.319 ± 0.055 | 1.609 ± 0.065 | 0.144 ± 0.060 | 1.310 ± 0.046 | 1.586 ± 0.064 | 0.148 ± 0.045 | 1.280 ± 0.039 | 1.595 ± 0.067 | 0.164 ± 0.057 | 1.273 ± 0.036 |
| CFR | 1.398 ± 0.032 | 0.084 ± 0.037 | 1.192 ± 0.019 | 1.431 ± 0.030 | 0.100 ± 0.055 | 1.207 ± 0.018 | 1.460 ± 0.036 | 0.090 ± 0.038 | 1.206 ± 0.021 | 1.476 ± 0.034 | 0.105 ± 0.043 | 1.207 ± 0.024 |
| GANITE | 1.449 ± 0.037 | 0.206 ± 0.165 | 1.307 ± 0.024 | 1.475 ± 0.026 | 0.211 ± 0.151 | 1.309 ± 0.019 | 1.503 ± 0.038 | 0.215 ± 0.141 | 1.305 ± 0.033 | 1.520 ± 0.055 | 0.188 ± 0.161 | 1.315 ± 0.022 |
| PSM | 2.164 ± 0.152 | 0.578 ± 0.232 | 1.658 ± 0.101 | 2.150 ± 0.110 | 0.403 ± 0.088 | 1.657 ± 0.093 | 2.165 ± 0.177 | 0.435 ± 0.243 | 1.662 ± 0.109 | 2.224 ± 0.186 | 0.459 ± 0.161 | 1.688 ± 0.120 |
| GOM | 1.629 ± 0.046 | 0.096 ± 0.048 | 1.225 ± 0.026 | 1.648 ± 0.040 | 0.084 ± 0.032 | 1.238 ± 0.022 | 1.673 ± 0.056 | 0.106 ± 0.030 | 1.243 ± 0.031 | 1.695 ± 0.045 | 0.098 ± 0.038 | 1.248 ± 0.028 |
| KOM | 1.629 ± 0.046 | 0.099 ± 0.046 | 1.225 ± 0.026 | 1.648 ± 0.039 | 0.086 ± 0.034 | 1.238 ± 0.021 | 1.672 ± 0.056 | 0.104 ± 0.027 | 1.243 ± 0.032 | 1.695 ± 0.045 | 0.100 ± 0.039 | 1.249 ± 0.028 |
| CEM | 1.499 ± 0.104 | 0.342 ± 0.121 | 1.333 ± 0.034 | 1.552 ± 0.063 | 0.433 ± 0.121 | 1.387 ± 0.039 | 1.570 ± 0.140 | 0.612 ± 0.158 | 1.383 ± 0.046 | 1.505 ± 0.258 | 0.638 ± 0.269 | 1.403 ± 0.056 |
| PM | 1.751 ± 0.185 | 0.404 ± 0.210 | 1.422 ± 0.125 | 1.777 ± 0.106 | 0.405 ± 0.090 | 1.431 ± 0.084 | 1.830 ± 0.192 | 0.449 ± 0.135 | 1.422 ± 0.117 | 1.763 ± 0.107 | 0.325 ± 0.094 | 1.385 ± 0.053 |
| CP | 1.394 ± 0.032 | 0.053 ± 0.022 | 1.191 ± 0.019 | 1.426 ± 0.028 | 0.049 ± 0.014 | 1.205 ± 0.017 | 1.457 ± 0.034 | 0.053 ± 0.016 | 1.206 ± 0.021 | 1.471 ± 0.032 | 0.055 ± 0.022 | 1.205 ± 0.023 |
| MitNet | 1.329 ± 0.024 | 0.142 ± 0.020 | 1.070 ± 0.015 | 1.356 ± 0.025 | 0.138 ± 0.018 | 1.089 ± 0.018 | 1.378 ± 0.022 | 0.138 ± 0.015 | 1.088 ± 0.013 | 1.388 ± 0.034 | 0.146 ± 0.024 | 1.087 ± 0.025 |
| MOGA | 1.316 ± 0.024 | 0.046 ± 0.018 | 1.063 ± 0.015 | 1.345 ± 0.024 | 0.045 ± 0.013 | 1.081 ± 0.017 | 1.368 ± 0.022 | 0.043 ± 0.019 | 1.082 ± 0.013 | 1.376 ± 0.031 | 0.064 ± 0.024 | 1.080 ± 0.023 |

**Evaluation Metrics** Following (Guo et al., 2023), we adopt multiple metrics to evaluate the performance of the conducted methods, including Precision in Estimation of Heterogeneous Effect (PEHE), Average Treatment Effect (ATE), and Average Mean Squared Error (AMSE). In particular, for the setting of multiple treatments, we consider the pair-wise version of ATE and PEHE denoted as mPEHE and mATE, respectively. The computational details of the metrics are presented in Appendix H.

### 4.2 RESULTS ON SEMI-SYNTHETIC AND SIMULATION DATA

In this section, we present the experimental results on both semi-synthetic and simulated datasets. More details about the dataset settings can be found in the Appendix I.

**Semi-synthetic data.** As shown in Table 1, in both binary treatments (News-2) and multiple treatments (News-4/8, TCGA), MOGA achieves promising performance and reliable results across different treatment scenarios. Specifically, compared to traditional matching methods such as PSM and kNN, MOGA achieves a significant improvement. This suggests that our approach effectively leverages information from all samples, leading to more accurate predictions of potential outcomes. In comparison to match-based methods like PM, MOGA shows superior performance. This is because MOGA simultaneously accounts for relations between neighbors and distance learning based on factual outcomes, which further enhances the quality of matching. In comparison to CP, which also employs a semi-supervised graph learning algorithm, MOGA demonstrates better performance. This can be attributed to the incorporation of outcome information during the distance learning in MOGA. Compared to other methods such as CFR and MitNet, MOGA also achieves highly competitive performance, which benefits from the usage of the underlying manifold structure of data and information from all groups to find close neighbors.

**Simulation data.** The results are shown in Table 2. To verify the robustness of different strengths of confounding bias, we progressively increase the mean differences between the groups to simulate different intensities of confounding bias. Overall, MOGA consistently outperforms other methods in terms of $\sqrt{\epsilon_{PEHE}}$, $\epsilon_{ATE}$ and $\sqrt{AMSE}$ with different levels of confounding biases, demonstrating superior effectiveness and stability. With the increase of strengths of confounding biases,

all methods perform worse, which is reasonable since confounding factors affect the performance of bias reduction and outcome prediction. Nevertheless, MOGA still achieves competitive performance compared with the others, which demonstrates the robustness of our method. Additionally, $m$ controls the distribution means, and the more spread out $m$ is, the less the group distributions overlap and the worse all methods perform. Our method can expand the matching pool by considering all groups rather than only the target group as candidate samples. As a result, our method still remains competitive with different values of $m$, which demonstrates the robustness of our method.

More experimental results can be found in Appendices J, K, and L, including visualization results, effects of distance functions and hyperparameters.

## 5 CONCLUSION

In this paper, we propose a matching method without group barriers for estimating heterogeneous treatment effects. Different from existing matching that finds neighbors from only the target group, our method considers neighbors from all the samples, so that closer samples can be matched to enhance counterfactual prediction. We analyze the estimation error of our matching method and propose a self optimal transport model based on our analysis. We further leverage the transport plan to design an outcome propagation method for counterfactual prediction, and incorporate factual outcomes to learn a distance as the transport cost. We conduct experiments on both binary and multiple treatment settings, and the experimental results demonstrate the effectiveness of our proposed method.

## 6 ACKNOWLEDGMENTS

This research was supported in part by National Science and Technology Major Project (2021ZD0111501), National Natural Science Foundation of China (62206061, U24A20233), National Science Fund for Excellent Young Scholars (62122022), Guangdong Basic and Applied Basic Research Foundation (2024A1515011901), Guangzhou Basic and Applied Basic Research Foundation (2023A04J1700), CCF-DiDi GAIA Collaborative Research Funds (CCF-DiDi GAIA 202521).

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

# A    RELATED WORKS

## A.1    CAUSAL EFFECT ESTIMATION

In the last decades, various methods based on machine learning have been proposed for causal effect estimation. Most of the existing methods do not consider unobserved confounders and can be categorized into three classes: reweighting, representation learning, and matching. The reweighting approach aims to construct pseudo-balanced groups by reweighting samples. Rosenbaum and Rubin (1983) estimate the propensity scores for samples and take the inverse of the propensity scores as the weights. To avoid estimating propensity scores, some methods learn sample weights to minimize the distribution shift between groups, in which the shift is measured by some predefined metric, such as the difference of moment (Hainmueller, 2012; Kuang et al., 2017) or the Integral Probability Metric (IPM) (Kong et al., 2023). The representation learning approach is devoted to learning balanced representations to reduce the distribution shift between groups (Johansson et al., 2016; Shi et al., 2019; Johansson et al., 2022). Shalit et al. (2017) trains a neural network to learn representations for minimizing the IPM between treated and control groups. Guo et al. (2023) instead leverages the mutual information to capture the distribution shift between groups in the setting of multiple treatments. Nevertheless, during the learning of balanced representations, some information highly related to potential outcomes could be lost, suffering from the over-balancing issue (Du et al., 2021; Yao et al., 2018).

There are also some studies considering unobserved confounding (Kallus et al., 2018; Wang et al., 2022), which poses difficulties for all the causal inference methods relying on standard assumptions. In this study, we still consider the standard assumption, including the unconfoundedness assumption. If the unconfoundedness assumption is violated, additional assumptions are required in existing studies, such as the presence of instrumental variables (Wang et al., 2022), or auxiliary random controlled trial data (Kallus et al., 2018).

Matching assumes that two samples with similar covariates usually have similar potential outcomes, based on this, to predict the counterfactual outcome of a treatment, classical matching finds the nearest neighbors in the target treatment group and predict counterfactual outcomes based on the matched neighbors (Li and Fu, 2017; Chang and Dy, 2017; Chu et al., 2020). The similarity between two samples is usually measured by the distance of covariates (Rubin, 1973) or the difference of propensity scores estimated by logistic regression (Rosenbaum and Rubin, 1983). Schwab et al. (2018) improve the matching approach based on propensity scores by learning a neural network, and propose a matching method within mini-batches. Kallus (2020) models matching as a problem of sample weight learning, and analyzes the estimation error under the framework of worst-case analysis.

Different from these matching methods that find matched samples in the target treatment group, we propose a novel matching method to find nearest neighbors from all the samples, so that more samples are involved to improve the data efficiency, and closer neighbors can be found to boost counterfactual prediction. We further model our method as a self optimal transport model, whose transport cost is supervised by factual outcomes and the solution is leveraged for counterfactual outcome prediction.

## A.2    OPTIMAL TRANSPORT

Optimal transport, which is originally proposed in (Monge, 1781) and then extended by Kantorovich in (Kantorovitch, 1958; Kantorovich, 2006), seeks the best plan to move one probability distribution into another distribution by minimizing the transport cost (Villani et al., 2009; Peyré et al., 2019). Recently, optimal transport has been widely applied in machine learning and data mining, including domain adaptation (Courty et al., 2017; Redko et al., 2017), generative model (Arjovsky et al., 2017; Tolstikhin et al., 2018), structured data analysis (Peyré et al., 2016; Titouan et al., 2019; Xu et al., 2019), *etc*. Optimal transport is also introduced into causal inference, focusing on confounding bias reduction between treated and control groups (Gunsilius and Xu, 2021; Wang et al., 2024; Dunipace, 2021). These methods usually learn weights or representations for samples to minimize the discrepancy measured by the theory of optimal transport (Li et al., 2021; Yan et al.). Different from them, we model our matching method as a self optimal transport model, which is able to find matched samples from all the groups rather than only the target group.

## B  CONSISTENCY

**Theorem 4.** *Under equation 7, assumption 4 and mild regularity conditions (Kong et al., 2023) (i.e., $n \to \infty$, $\sum_{j=1}^{n} \mathbb{E}[W_{ij}^2] = 0$), while $n \to \infty$, the outcome estimation error $|f_t(x_i) - \hat{Y}_t(x_i)| \to 0$.*

*Proof.* We analyze the outcome estimation error as follows

$$
\begin{aligned}
|f_t(x_i) - \hat{Y}_t(x_i)| &= |f_t(x_i) - \sum_{j=1}^{n} W_{ij} Y_t(x_j)| \\
&= |f_t(x_i) - \sum_{j=1}^{n} W_{ij}(f_t(x_j) + \xi_j)| \\
&= |\sum_{j=1}^{n} W_{ij}(f_t(x_i) - f_t(x_j)) - \sum_{j=1}^{n} W_{ij}\xi_j| \\
&\leq \sum_{j=1}^{n} W_{ij}|f_t(x_i) - f_t(x_j)| + |\sum_{j=1}^{n} W_{ij}\xi_j| \\
&\leq L \sum_{j=1}^{n} W_{ij}\|\phi(x_i) - \phi(x_j)\| + |\sum_{j=1}^{n} W_{ij}\xi_j|.
\end{aligned}
\tag{23}
$$

Since the weights $W_{ij}$ are obtained via self optimal transport learning, where a larger $\|\phi(\mathbf{x}_i) - \phi(\mathbf{x}_j)\|$ corresponds to a smaller $W_{ij}$, it follows that for sufficiently large $n$, the product $W_{ij}\|\phi(\mathbf{x}_i) - \phi(\mathbf{x}_j)\|$ approaches zero (Kong et al., 2023). We next show that as $n \to \infty$, the sum $\sum_{j=1}^{n} W_{ij}\xi_j$ converges to zero under the regularity condition (Kong et al., 2023), i.e., $n \to \infty$, $\sum_{j=1}^{n} \mathbb{E}[W_{ij}^2] = 0$. We first show that as $n \to \infty$, its mean is 0:

$$
\mathbb{E}[\sum_{j=1}^{n} W_{ij}\xi_j] = \sum_{j=1}^{n} \mathbb{E}[W_{ij}]\mathbb{E}[\xi_j] = \sum_{j=1}^{n} \mathbb{E}[W_{ij}] \times 0 = 0,
\tag{24}
$$

where the first equality is based on $W_{ij} \perp\!\!\!\perp \xi_j$, and also its variance is zero:

$$
\begin{aligned}
Var[\sum_{j=1}^{n} W_{ij}\xi_j] &= \sum_{j=1}^{n} Var[W_{ij}\xi_j] + \sum_{k \neq j} Cov(W_{ij}\xi_j, W_{ik}\xi_k) \\
&= \sum_{j=1}^{n} Var[W_{ij}\xi_j] + \sum_{k \neq j} (\mathbb{E}[W_{ij}\xi_j W_{ik}\xi_k] - \mathbb{E}[W_{ij}\xi_j]\mathbb{E}[W_{ik}\xi_k)]) \\
&= \sum_{j=1}^{n} Var[W_{ij}\xi_j] + 0 - 0 \\
&= \sum_{j=1}^{n} \mathbb{E}[W_{ij}^2] Var[\xi_j] \\
&= \sigma^2 \sum_{j=1}^{n} \mathbb{E}[W_{ij}^2] \\
&= 0,
\end{aligned}
\tag{25}
$$

where the third equality is based on zero mean of $\xi_j$ and $\{W_{ij}, W_{ik}\} \perp\!\!\!\perp \xi_j$ and $\xi_j \perp\!\!\!\perp \xi_k$, and in fifth equality we set $\sigma^2 = Var[\xi_j]$, and the last equality holds due to the regularity condition. Eq. 24 and Eq. 25 together imply $n \to \infty$, $\sum_{j=1}^{n} W_{ij}\xi_j \to 0$, which finishes the proof.

$\square$

## C  PROOF OF THEOREM 1

**Theorem 1** *Let $\epsilon_{Y_t} = \sum_{i=1}^{n} \mathbb{E}(f_t(\mathbf{x}_i) - \hat{Y}_t(\mathbf{x}_i))^2$ be the estimation error of the potential outcomes under the treatment $t$, if the assumption $\forall i, Var(y_i|\mathbf{x}_i, t_i) = \eta^2$ holds, then the error is upper bounded by the following:*

$$\epsilon_{Y_t} \leq 2L_t^2 \sum_{i=1}^{n} \sum_{j=1}^{n} W_{ij} \|\phi(\mathbf{x}_i) - \phi(\mathbf{x}_j)\|_2^2 + 2n\eta^2 \sum_{i=1}^{n} \sum_{j=1}^{n} W_{ij}^2. \tag{26}$$

*Proof.* Based on the assumptions in Section 2.2 and the condition $\sum_{j=1}^{n} W_{ij} = 1$, the upper bound is derived as follows:

$$\mathbb{E}(f_t(\mathbf{x}_i) - \hat{Y}_t(\mathbf{x}_i))^2 = \mathbb{E}(f_t(\mathbf{x}_i) - \sum_{j=1}^{n} W_{ij}(f_t(\mathbf{x}_j) + \xi_j))^2$$

$$= \mathbb{E}(\sum_{j=1}^{n} W_{ij}(f_t(\mathbf{x}_i) - f_t(\mathbf{x}_j)) + \sum_{j=1}^{n} W_{ij}\xi_j)^2$$

$$\leq 2(\sum_{j=1}^{n} W_{ij}(f_t(\mathbf{x}_i) - f_t(\mathbf{x}_j)))^2 + 2\mathbb{E}(\sum_{j=1}^{n} W_{ij}\xi_j)^2, \tag{27}$$

where the inequality holds because of the condition that $(a+b)^2 \leq 2a^2 + 2b^2$. In the following, we analyze the two terms $(\sum_{j=1}^{n} W_{ij}(f_t(\mathbf{x}_i) - f_t(\mathbf{x}_j)))^2$ and $2\mathbb{E}(\sum_{j=1}^{n} W_{ij}\xi_j)^2$, respectively.

For the first term in (27), we have

$$2(\sum_{j=1}^{n} W_{ij}(f_t(\mathbf{x}_i) - f_t(\mathbf{x}_j)))^2 = 2(\sum_{j=1}^{n} \sqrt{W_{ij}}\sqrt{W_{ij}}(f_t(\mathbf{x}_i) - f_t(\mathbf{x}_j)))^2$$

$$\leq 2(\sum_{j=1}^{n} W_{ij})(\sum_{j=1}^{n} W_{ij}(f_t(\mathbf{x}_i) - f_t(\mathbf{x}_j))^2)$$

$$= 2\sum_{j=1}^{n} W_{ij}(f_t(\mathbf{x}_i) - f_t(\mathbf{x}_j))^2$$

$$\leq 2L_t^2 \sum_{j=1}^{n} W_{ij} \|\phi(\mathbf{x}_i) - \phi(\mathbf{x}_j)\|_2^2, \tag{28}$$

where the first inequality holds according to the Cauchy–Schwarz inequality, the second inequality holds because of the Lipschitz continuity of the function $f_t(\cdot)$.

For the second term in (27), we have

$$2\mathbb{E}(\sum_{j=1}^{n} W_{ij}\xi_j)^2 \leq 2(\sum_{j=1}^{n} W_{ij}^2)\mathbb{E}(\sum_{j=1}^{n} \xi_j^2)$$

$$= 2n\eta^2 \sum_{j=1}^{n} W_{ij}^2, \tag{29}$$

where according to the Cauchy–Schwarz inequality, then can simply rewrite $\mathbb{E}(\sum_{j=1}^{n} \xi_j^2)$ as $2(n-1)\eta^2$.

Based on the conclusions above, we can derive:

$$(f_t(\mathbf{x}_i) - \hat{Y}_t(\mathbf{x}_i))^2 \leq 2L_t^2 \sum_{j=1}^{n} W_{ij} \|\phi(\mathbf{x}_i) - \phi(\mathbf{x}_j)\|_2^2 + 2n\eta^2 \sum_{j=1}^{n} W_{ij}^2, \tag{30}$$

and Theorem 1 can be obtained by considering all the samples. □

## D  PROOF OF THEOREM 2

**Theorem 2** *Let $\hat{Y}_t(\mathbf{x}_i)$ denote the predicted outcome for the $i$-th sample under the treatment $t$. The effect estimation error is measured by the pairwise precision in estimation of heterogeneous effect (mPEHE) is defined as $\epsilon_{mPEHE} = \frac{2}{nT(T+1)} \sum_{0 \leq t' < t \leq T} \sum_{i=1}^{n} ((\hat{Y}_t(\mathbf{x}_i) - \hat{Y}_{t'}(\mathbf{x}_i)) - (f_t(\mathbf{x}_i) - f_{t'}(\mathbf{x}_i)))^2$ (Schwab et al., 2018; Guo et al., 2023). The effect estimation error is upper bounded by the outcome estimation error as follows*

$$\epsilon_{mPEHE} \leq \frac{4}{n(T+1)} \sum_{t=0}^{T} \epsilon_{Y_t}. \tag{31}$$

*Proof.*

$$
\begin{aligned}
\epsilon_{mPEHE} &= \frac{2}{nT(T+1)} \sum_{0 \leq t' < t \leq T} \sum_{i=1}^{n} \left( (\hat{Y}_t(\mathbf{x}_i) - \hat{Y}_{t'}(\mathbf{x}_i)) - (f_t(\mathbf{x}_i) - f_{t'}(\mathbf{x}_i)) \right)^2 \\
&= \frac{2}{nT(T+1)} \sum_{0 \leq t' < t \leq T} \sum_{i=1}^{n} \left[ \left( (\hat{Y}_t(\mathbf{x}_i) - f_t(\mathbf{x}_i)) + (f_{t'}(\mathbf{x}_i) - \hat{Y}_{t'}(\mathbf{x}_i)) \right)^2 \right] \\
&\leq \frac{4}{nT(T+1)} \sum_{0 \leq t' < t \leq T} \sum_{i=1}^{n} \left[ \left( (\hat{Y}_t(\mathbf{x}_i) - f_t(\mathbf{x}_i))^2 + (\hat{Y}_{t'}(\mathbf{x}_i) - f_{t'}(\mathbf{x}_i))^2 \right) \right] \\
&= \frac{4}{n(T+1)} \sum_{t=0}^{T} \sum_{i=1}^{n} \left[ (\hat{Y}_t(\mathbf{x}_i) - f_t(\mathbf{x}_i))^2 \right]. \\
&= \frac{4}{n(T+1)} \sum_{t=0}^{T} \epsilon_{Y_t}. \qquad \square
\end{aligned}
$$

## E  THEOREMS REGARDING AMSE AND ATE

**Theorem 5.** *Let $AMSE = \frac{1}{n(T+1)} \sum_{t=0}^{T} \sum_{i=1}^{n} (\hat{Y}_t(\mathbf{x}_i) - f_t(\mathbf{x}_i))^2$ be the average mean squared error of the potential outcomes. It is upper bounded by the following*

$$AMSE \leq \frac{2}{n(T+1)} \sum_{t=0}^{T} \sum_{i=1}^{n} (L_t^2 \sum_{j=1}^{n} W_{ij} \|\phi(\mathbf{x}_i) - \phi(\mathbf{x}_j)\|_2^2 + n\eta^2 \sum_{j=1}^{n} W_{ij}^2). \tag{32}$$

*Proof.*

$$AMSE = \frac{1}{n(T+1)} \sum_{t=0}^{T} \sum_{i=1}^{n} (\hat{Y}_t(\mathbf{x}_i) - f_t(\mathbf{x}_i))^2 = \frac{1}{n(T+1)} \sum_{t=0}^{T} \epsilon_{Y_t}, \tag{33}$$

By combining Theorem 1, it follows directly that:

$$AMSE \leq \frac{2}{n(T+1)} \sum_{t=0}^{T} \sum_{i=1}^{n} (L_t^2 \sum_{j=1}^{n} W_{ij} \|\phi(\mathbf{x}_i) - \phi(\mathbf{x}_j)\|_2^2 + n\eta^2 \sum_{j=1}^{n} W_{ij}^2). \qquad \square$$

**Theorem 6.** *Let $\epsilon_{mATE} = \frac{2}{T(T+1)} \sum_{0 \leq t' < t \leq T} |\frac{1}{n} \sum_{i=1}^{n} (\hat{Y}_t(\mathbf{x}_i) - \hat{Y}_{t'}(\mathbf{x}_i)) - \frac{1}{n} \sum_{i=1}^{n} (f_t(\mathbf{x}_i) - f_{t'}(\mathbf{x}_i))|$ be the error of the average treatment effect. It is upper bounded by the following*

$$\epsilon_{mATE} \leq \frac{2}{n(T+1)} \sum_{t=0}^{T} \sum_{i=1}^{n} (L \sum_{j=1}^{n} W_{ij} |\phi(\mathbf{x}_j) - \phi(\mathbf{x}_i)| + n\eta \sum_{j=1}^{n} |W_{ij}|). \tag{34}$$

*Proof.*

$$\epsilon_{mATE} = \frac{2}{T(T+1)} \sum_{0 \leq t' < t \leq T} |\frac{1}{n} \sum_{i=1}^{n} (\hat{Y}_t(\mathbf{x}_i) - \hat{Y}_{t'}(\mathbf{x}_i)) - \frac{1}{n} \sum_{i=1}^{n} (f_t(\mathbf{x}_i) - f_{t'}(\mathbf{x}_i))|$$

$$= \frac{2}{T(T+1)} \sum_{0 \leq t' < t \leq T} |\frac{1}{n} \sum_{i=1}^{n} ((\hat{Y}_t(\mathbf{x}_i) - \hat{Y}_{t'}(\mathbf{x}_i)) - (f_t(\mathbf{x}_i) - f_{t'}(\mathbf{x}_i)))|$$

$$\leq \frac{2}{nT(T+1)} \sum_{0 \leq t' < t \leq T} \sum_{i=1}^{n} (|\hat{Y}_t(\mathbf{x}_i) - f_t(\mathbf{x}_i)| + |\hat{Y}_{t'}(\mathbf{x}_i) - f_{t'}(\mathbf{x}_i)|)$$

$$= \frac{2}{n(T+1)} \sum_{t=0}^{T} \sum_{i=1}^{n} |\hat{Y}_t(\mathbf{x}_i) - f_t(\mathbf{x}_i)|$$

$$= \frac{2}{n(T+1)} \sum_{t=0}^{T} \sum_{i=1}^{n} |\sum_{j=1}^{n} W_{ij}(f_t(\mathbf{x}_j) + \xi_j) - \sum_{j=1}^{n} W_{ij} f_t(\mathbf{x}_i)|$$

$$= \frac{2}{n(T+1)} \sum_{t=0}^{T} \sum_{i=1}^{n} |\sum_{j=1}^{n} W_{ij}(f_t(\mathbf{x}_j) - f_t(\mathbf{x}_i) + \xi_j)|$$

$$\leq \frac{2}{n(T+1)} \sum_{t=0}^{T} \sum_{i=1}^{n} (\sum_{j=1}^{n} W_{ij}|f_t(\mathbf{x}_j) - f_t(\mathbf{x}_i)| + |\sum_{j=1}^{n} W_{ij}\xi_j|)$$

$$\leq \frac{2}{n(T+1)} \sum_{t=0}^{T} \sum_{i=1}^{n} (L \sum_{j=1}^{n} W_{ij}|\phi(\mathbf{x}_j) - \phi(\mathbf{x}_i)| + n\eta \sum_{j=1}^{n} |W_{ij}|). \qquad \square$$

## F  PROOF OF PROPOSITION 3

**Proposition 3** *Let $\mathbf{X}_t \in \mathbb{R}^{n_t \times d}$ be the matrix including all the samples in the treatment group $t$. Problem 20 is equivalent to the following problem*

$$\min_{\mathbf{P}} \ \mathrm{tr}\left(\mathbf{P}^\top (\sum_{t=0}^{T} \mathbf{\Theta}_t)\mathbf{P}\right) \quad s.t. \ \mathbf{P}^\top \mathbf{P} = \mathbf{I}, \tag{35}$$

*where the matrix $\mathbf{\Theta}_t$ is constructed as*

$$\mathbf{\Theta}_t = 2(\mathbf{X}_t)^\top diag(\boldsymbol{\gamma}_t \mathbf{1} - \boldsymbol{\gamma}_t)\mathbf{X}_t. \tag{36}$$

*The closed-form solution to this problem is obtained by the eigenvectors associated with the $d'$ smallest eigenvalues of the matrix $\sum_{t=0}^{T} \mathbf{\Theta}_t$.*

*Proof.* First of all, we implement $\phi(\cdot)$ as a projection operation $\phi(\mathbf{x}) = \mathbf{P}^\top \mathbf{x}$ with $\mathbf{P} \in \mathbb{R}^{d \times d'}$ are the parameters to be optimized. Beside, we set $C_{t;ij}^{\mathbf{P}}$ denotes $\|\mathbf{P}^\top \mathbf{x}_i - \mathbf{P}^\top \mathbf{x}_j\|_2^2$. After that, the transport cost between $\mathbf{x}_i$ and $\mathbf{x}_j$ can be rewritten as:

$$\langle \mathbf{C}_t^{\mathbf{P}}, \boldsymbol{\gamma}_t \rangle = \sum_{i=1}^{n_t} \sum_{j=1}^{n_t} C_{t;ij}^{\mathbf{P}} \gamma_{t;ij}$$

$$= \sum_{i=1}^{n_t} \sum_{j=1}^{n_t} \|\mathbf{P}^\top \mathbf{x}_i - \mathbf{P}^\top \mathbf{x}_j\|_2^2 \gamma_{t;ij}$$

$$= 2\sum_{i=1}^{n_t} (\|\mathbf{P}^\top \mathbf{x}_i\|_2^2) - 2\sum_{i=1}^{n_t} \sum_{j=1}^{n_t} (\langle \mathbf{P}^\top \mathbf{x}_i, \mathbf{P}^\top \mathbf{x}_j \rangle) \gamma_{t;ij}$$

$$= 2\langle (\mathbf{X}_t\mathbf{P})(\mathbf{X}_t\mathbf{P})^\top, diag(\boldsymbol{\gamma}\mathbf{1}) \rangle - 2\langle (\mathbf{X}_t\mathbf{P})(\mathbf{X}_t\mathbf{P})^\top, \boldsymbol{\gamma}_t \rangle$$

$$= 2\,\mathrm{tr}(\mathbf{P}^\top \mathbf{X}_t^\top (diag(\boldsymbol{\gamma}\mathbf{1}) - \boldsymbol{\gamma})\mathbf{X}_t\mathbf{P})$$

$$= \mathrm{tr}(\mathbf{P}^\top \mathbf{\Theta}_t \mathbf{P}). \tag{37}$$

Based on this, the objective function of Problem (20) can be written as

$$\sum_{t=0}^{T}\langle \mathbf{C}_t^{\mathbf{P}}, \boldsymbol{\gamma}_t \rangle = \sum_{t=0}^{T} \mathrm{tr}\left(\mathbf{P}^\top \boldsymbol{\Theta}_t \mathbf{P}\right) = \mathrm{tr}\left(\mathbf{P}^\top (\sum_{t=0}^{T} \boldsymbol{\Theta}_t)\mathbf{P}\right). \tag{38}$$

The solution to minimize this objective is the eigenvectors associated with the $d'$ smallest eigenvalues of the matrix $\sum_{t=0}^{T} \boldsymbol{\Theta}_t$. □

## G   PSEUDO-CODE OF MATCHING WITHOUT GROUP BARRIER (MOGA).

Algorithm 1 presents the pseudo-code of our method MOGA.

---
**Algorithm 1** Matching without Group Barrier (MOGA)
:
---
**Input:** Data samples $\{(\mathbf{x}_i, y_i, t_i)\}_{i=1}^n$.
1: Initialize $\boldsymbol{\gamma}_t$ by solving Problem (15).
2: **loop**
3:     Update $\mathbf{P}$ according to Proposition 3.
4:     Update $\boldsymbol{\gamma}_t$ by solving Problem (18).
5: **end loop**
6: Construct the cost matrix $\mathbf{C}^\phi$ based on Eq. (19).
7: Obtain matching matrix $\boldsymbol{\gamma}$ by solving Problem (11).
8: **loop**
9:     Update outcome matrix based on Eq. (14).
10: **end loop**
---

For Algorithm 1, let $n$ and $n_t$ be the numbers of all the samples and the samples in the treatment group $t$, $d$ and $d'$ be the numbers of the features before and after projection. For distance learning, the complexity of Step 3 is $O(n_t^2 d + n_t d^2 + d^3)$, the complexity of Step 4 is $O(n_t^2 d')$. For optimal transport matching, the complexity of Steps 6 and 7 is $O(ndd' + n^2 d')$. For counterfactual prediction, the complexity of Step 9 is $O(n^2 T)$, where $T$ is the number of different treatment values.

The overall space complexity of the Algorithm 1 is dominated by the storage of the $N \times N$ transition probability matrix $\mathbf{W}$, which involves two steps: optimal transport and label propagation. First, during the calculation of optimal transport, the primary memory requirement is for simultaneously storing the input data $X$ and the probability matrix $\mathbf{W}$, whose complexities are $O(Nd)$ and $O(N^2)$, respectively, where $d$ is the feature dimension. Second, the total space required during label propagation accommodates the largest input matrix $\mathbf{W} \in \mathbb{R}^{N \times N}$ along with the label matrix $\mathbf{Y} \in \{0,1\}^{N \times T}$, whose complexities are $O(N^2)$ and $O(NT)$, respectively, where $T \ll N$. In summary, the algorithm's dominant space complexity is $O(N^2 + Nd)$.

## H   EVALUATION METRICS

To evaluate the performance of the conducted methods, we follow (Schwab et al., 2018) to adopt the following metrics

$$\epsilon_{mPEHE} = \frac{2}{nT(T+1)} \sum_{0 \leq t' < t \leq T} \sum_{i=1}^{n} ((\hat{Y}_t(\mathbf{x}_i) - \hat{Y}_{t'}(\mathbf{x}_i)) - (f_t(\mathbf{x}_i) - f_{t'}(\mathbf{x}_i)))^2, \tag{39}$$

$$\epsilon_{mATE} = \frac{2}{T(T+1)} \sum_{0 \leq t' < t \leq T} |\frac{1}{n} \sum_{i=1}^{n} (\hat{Y}_t(\mathbf{x}_i) - \hat{Y}_{t'}(\mathbf{x}_i)) - \frac{1}{n} \sum_{i=1}^{n} (f_t(\mathbf{x}_i) - f_{t'}(\mathbf{x}_i))|. \tag{40}$$

Besides, we also add a metric:

$$AMSE = \frac{1}{n(T+1)} \sum_{t=0}^{T} \sum_{i=1}^{n} (\hat{Y}_t(\mathbf{x}_i) - f_t(\mathbf{x}_i))^2. \tag{41}$$

# I   DATASET SETTING

**News**   The News dataset is first proposed as a benchmark for counterfactual inference by Johansson et al. (2016) and is used in the multiple treatment setting in Schwab et al. (2018). The News dataset simulates counterfactual inference by modeling news articles as topic distributions $z(\mathbf{x})$, derived from a topic model trained on the NY Times corpus. Multiple centroids are randomly chosen in the topic space, where one centroid represents the control group while the other centroids represent treated groups viewing devices (treatments). Each centroid $z_j$ is associated with a Gaussian outcome distribution: $m_j \sim \mathcal{N}(0.45, 0.15)$, $\sigma_j \sim \mathcal{N}(0.1, 0.05)$, from which ideal potential outcomes are sampled as $\tilde{y}_j \sim \mathcal{N}(m_j, \sigma_j) + \epsilon$, where $\epsilon \sim \mathcal{N}(0, 0.15)$. The unscaled potential outcomes are computed as $\bar{y}_j = \tilde{y}_j \cdot [D(z(\mathbf{x}), z_j) + D(z(\mathbf{x}), z_c)]$, where $D(\cdot, \cdot)$ is the Euclidean distance, and $z_c$ represents the control centroid. The treatment assignment follows $t|x \sim \text{Bernoulli}(\text{softmax}(\nu \bar{y}_j))$, with $\nu$ controlling the strength of assignment bias ($\nu = 0$ implies no bias). The true observed outcomes are scaled by a constant $D = 50$: $y_j = D \cdot \bar{y}_j$. The dataset can simulate $k = 2, 4, 8$ treatments with $\nu = 10$, enabling flexible modeling of counterfactual inference scenarios.

**TCGA**   The Cancer Genome Atlas (TCGA) project collects gene expression data from 9,659 individuals with various types of cancer, incorporating 20,531 covariates Weinstein et al. (2013). The dataset includes three clinical treatment options: medication, chemotherapy, and surgery. To estimate the risk of cancer recurrence following any of these treatments, a synthetic outcome function, the dose-response curve, was applied using real-world gene expression data. The modeling of outcomes follows the method described by Schwab et al. (2020), where the treatment assignment bias coefficient is set to $\nu = 10$. Furthermore, to evaluate the robustness of the model, we artificially introduce Gaussian noise to the outcome variable to simulate random perturbations in the experiment. Specifically, for given sample $i$, the observed outcome $y_i = y_{i,t_i} + \xi_i$ where $\xi_i \sim \mathcal{N}(0, \sigma^2)$ and we set $\sigma = 5$.

**Simulation data**   Following (Yao et al., 2018; Hatt and Feuerriegel, 2021), we generate synthetic data for four treated group and one control groups by sampling features $\mathbf{x}$ from Gaussian mixture distribution $\mathbf{x} \sim w_1 \cdot \mathcal{N}(\boldsymbol{m}, \boldsymbol{\Sigma}) + w_2 \cdot \mathcal{N}(2\boldsymbol{m}, \boldsymbol{\Sigma})$, where $\boldsymbol{m} = [m, \ldots, m] \in \mathbb{R}^d$ and $\boldsymbol{\Sigma} = 0.5 \cdot (\Sigma_{\text{rand}} \cdot \Sigma_{\text{rand}}^\top)$, with $\Sigma_{\text{rand}} \sim \mathcal{U}((0, \text{bound})^{d \times d}$. The outcomes $y$ are modeled as $y = \sin(\mathbf{w}_1^\top \mathbf{x}) \cdot \exp(\cos(\mathbf{w}_2^\top (\mathbf{x} \odot \mathbf{x}))) + \xi$, where $\mathbf{w}_1, \mathbf{w}_2 \sim \mathcal{U}((0, 1)^{d \times k})$ are random weight metrices, $k$ represents the total number of treatments and control groups, and $\xi \sim \mathcal{N}(0, 0.5)$ represents noise. We vary the value of $\mathbf{m}$ across different groups to explore the performance of the conducted methods under different data conditions.

# J   MATCHING VISUALIZATION

We conduct an experiment on simulation data to visualize the matching results of our method. The data consist of three treated groups and one control group, each of which includes with 20 samples with 25 features. Similar to simulation data generation in Section I, we generate synthetic data as follows. Let $\mathbf{x} \sim w_1 \cdot \mathcal{N}(\mathbf{m}, 0.5\boldsymbol{\Sigma}_{\text{rand}}\boldsymbol{\Sigma}_{\text{rand}}^\top) + w_2 \cdot \mathcal{N}(2\mathbf{m}, 0.5\boldsymbol{\Sigma}_{\text{rand}}\boldsymbol{\Sigma}_{\text{rand}}^\top)$ where $\mathbf{m} = [1.5, 1.75, 1.0, 0.75]^\top$ and $\boldsymbol{\Sigma}_{\text{rand}} \sim \mathcal{U}((0, [1.2, 3.4, 2.6, 0.8]^\top)^{d \times d})$. We learn the projection matrix $\mathbf{P}$ to map data into a 2D space, and show the matching results in Figure 1. We find the nearest neighbors for the objective samples, the dotted lines represent matched samples, and the color depth of matched samples represents the matching degree. We observe that samples in different groups are matched, and closer neighbors have darker colors, which means that they contribute more to the prediction.

# K   MORE RESULTS

## K.1   DIFFERENT DISTANCE LEARNING

Besides the distance learned in Section 3.3, we also calculate the distance between $\mathbf{x}_i$ and $\mathbf{x}_j$ using the following methods: For the squared Euclidean distance, we have $c_\phi(\mathbf{x}_i, \mathbf{x}_j) = \|\mathbf{x}_i - \mathbf{x}_j\|_2^2$. For the Cosine distance, we measure the distance without considering the scale of covariates, which is given as $c_\phi(\mathbf{x}_i, \mathbf{x}_j) = \|\frac{\mathbf{x}_i}{\|\mathbf{x}_i\|_2} - \frac{\mathbf{x}_i}{\|\mathbf{x}_i\|_2}\|_2^2 = 2 - 2\frac{\mathbf{x}_i \cdot \mathbf{x}_j}{\|\mathbf{x}_i\|_2 \|\mathbf{x}_j\|_2}$. We take the TCGA dataset as an

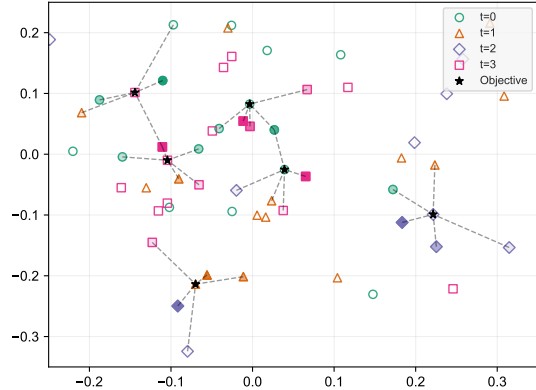

Figure 1: Visualization results of our matching method MOGA. Hollow points in different colors and shapes represent different groups, while solid black stars denote the objective nodes. The figure displays the five matched samples with the top matching degrees, with darker colors indicating higher matching degrees.

example to compare different distances used in our method, and report the results in Table 3. We observe that MOGA takes the outcome into consideration through optimal transport, thus achieving the best performance.

Table 3: Results of different distances on TCGA dataset.

|  | $\sqrt{\epsilon_{PEHE}}$ | $\epsilon_{ATE}$ | $\sqrt{AMSE}$ |
|---|---|---|---|
| Euclidean | 12.5989± 0.0263 | 6.0822± 0.0419 | 9.7808± 0.0203 |
| Cosine distance | 11.1576± 0.0452 | 4.1404± 0.0892 | 8.3178± 0.0416 |
| MOGA | 10.5965± 0.0263 | 2.7850± 0.0752 | 7.7511± 0.0407 |

## K.2 COMPARISON WITH TRADITIONAL MATCHING

We also consider a variant of our matching method, which leverages the distance learned in Section 3.3 but selecting neighbors within the target group only. We take the News-4 dataset as an example and report the results in Table 4. We observe that MOGA performs better which demonstrates the advantage of our matching method considering all the samples regardless of their received treatments.

Table 4: Results of different matching methods on the News-4 dataset.

|  | $\sqrt{\epsilon_{PEHE}}$ | $\epsilon_{ATE}$ | $\sqrt{AMSE}$ |
|---|---|---|---|
| matching only within the target group | $8.0039 \pm 2.2043$ | $2.9041 \pm 1.7250$ | $8.0388 \pm 2.5587$ |
| MOGA | $5.9601 \pm 1.1798$ | $1.1551 \pm 0.7063$ | $4.4197 \pm 0.9348$ |

## K.3 PERFORMANCE OF REAL-WORLD DATASET

To compare the performance in the Real-world dataset, we report the results of $\epsilon_{ATE}$ on the Lalonde dataset(LaLonde, 1986) in Table 5. The LaLonde dataset consists of experimental and observational components. The experimental part comes from the National Supported Work (NSW) randomized controlled trial, while the control group is replaced by observational data from the Panel Study of

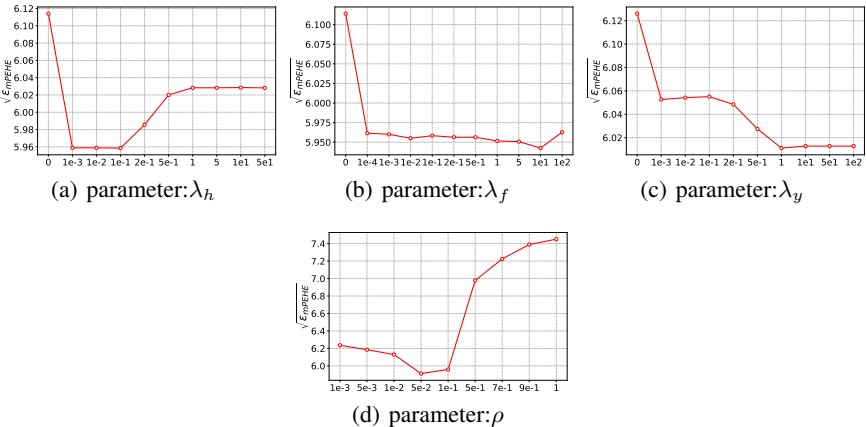

Figure 2: Results of varying values of hyperparameters on News-4 dataset.

Table 5: MAE results on Lalonde dataset (mean $\pm$ standard deviation). Lower is better.

| Method | MAE of Lalonde |
|--------|----------------|
| MitNet | $393.9513 \pm 87.6213$ |
| CFR | $495.3089 \pm 59.5542$ |
| PSM | $728.7922 \pm 500.1740$ |
| kNN | $275.2712 \pm 148.3843$ |
| GOM | $434.4803 \pm 231.6803$ |
| KOM | $438.3496 \pm 255.4207$ |
| CEM | $796.7605 \pm 610.0612$ |
| MOGA | $246.1526 \pm 86.9842$ |

Income Dynamics (PSID) dataset. The treatment indicates participation in a job training program, and the outcome is earnings in 1978. The dataset is widely used for benchmarking causal inference methods. We observe that our proposed method still achieves promising results.

### K.4 MORE BASELINES

We also compare with more baselines on the News data. **S-learner** is a single unified model trained with the treatment indicator as an input feature to directly estimate potential outcomes under different treatments and their difference. **T-learner** uses two separate models trained for the treated and control groups, and the individual treatment effect is obtained by taking the difference between their predictions. **X-learner** (Künzel et al., 2019) estimates ITE via cross-fitting of imputed treatment effects and propensity score–based weighting, making it especially effective under treatment–control imbalance. **R-learner** (Nie and Wager, 2021) formulates ITE estimation through orthogonalized residual regression by removing the effects of covariates on both the outcome and treatment, yielding robustness to confounding. **DragonNet** (Shi et al., 2019) leverages propensity scores and targeted regularization to improve outcome prediction and stabilize treatment effect estimation. **CE-RCFR** (Wang et al., 2022) enhances ITE estimation through relaxed optimal transport alignment and consensus aggregation, simultaneously mitigating mini-batch noise and gradient conflicts. **CBPS** (Imai and Ratkovic, 2014) estimates propensity scores while directly optimizing covariate balance between treatment groups, improving causal effect estimation. **MALTS** (Parikh et al., 2022) is a matching method that leverages a learnable distance metric to optimize feature-space similarity and accurately estimate ITE. The results are listed in Table 6. We observe that our method achieves promising performance

Table 6: Results on News dataset (mean ± standard deviation) on more baselines. Lower is better.

| Dataset | News-2 | | | News-4 | | | News-8 | | |
|---|---|---|---|---|---|---|---|---|---|
| Metric | $\sqrt{\epsilon_{PEHE}}$ | $\hat{\epsilon}_{ATE}$ | $\sqrt{AMSE}$ | $\sqrt{\hat{\epsilon}_{mPEHE}}$ | $\hat{\epsilon}_{mATE}$ | $\sqrt{AMSE}$ | $\sqrt{\hat{\epsilon}_{mPEHE}}$ | $\hat{\epsilon}_{mATE}$ | $\sqrt{AMSE}$ |
| DragonNet | 9.408 ± 2.324 | 1.990 ± 1.567 | 6.969 ± 1.838 | 10.359 ± 4.153 | 3.787 ± 3.960 | 8.015 ± 2.611 | 16.904 ± 5.637 | 8.503 ± 7.913 | 18.367 ± 6.896 |
| CE-RCFR | 9.316 ± 2.397 | 1.613 ± 1.561 | 6.972 ± 1.836 | 9.458 ± 1.828 | 2.791 ± 1.179 | 8.017 ± 1.736 | 10.545 ± 1.429 | 4.057 ± 1.172 | 10.506 ± 1.855 |
| R-learner | 9.336 ± 2.367 | 1.710 ± 1.521 | 6.902 ± 2.276 | 9.646 ± 2.048 | 3.118 ± 1.654 | 8.479 ± 2.212 | 10.470 ± 1.282 | 3.867 ± 0.908 | 11.035 ± 1.743 |
| S-learner | 8.646 ± 2.374 | 1.533 ± 1.468 | 6.386 ± 1.743 | 8.993 ± 1.846 | 2.591 ± 1.282 | 7.845 ± 1.747 | 10.320 ± 1.313 | 3.900 ± 0.951 | 10.577 ± 1.670 |
| T-learner | 8.381 ± 2.246 | 1.581 ± 1.512 | 6.258 ± 1.723 | 8.800 ± 1.846 | 2.658 ± 1.273 | 7.774 ± 1.747 | 10.284 ± 1.297 | 3.914 ± 0.927 | 10.582 ± 1.645 |
| X-learner | 8.577 ± 2.270 | 1.574 ± 1.510 | 6.329 ± 1.732 | 8.825 ± 1.842 | 2.661 ± 1.275 | 7.789 ± 1.748 | 10.250 ± 1.293 | 3.922 ± 0.925 | 10.570 ± 1.643 |
| CBPS | 10.646 ± 2.847 | 2.872 ± 2.439 | 7.528 ± 2.013 | 13.214 ± 2.679 | 2.891 ± 1.763 | 10.929 ± 2.679 | 15.817 ± 1.987 | 3.437 ± 0.974 | 14.845 ± 2.385 |
| MALTS | 10.692 ± 2.663 | 1.553 ± 1.486 | 7.911 ± 2.025 | 10.382 ± 1.996 | 2.658 ± 1.273 | 8.713 ± 1.833 | 13.123 ± 1.890 | 3.278 ± 1.439 | 10.224 ± 2.435 |
| **MOGA** | **5.081 ± 1.693** | **0.449 ± 0.344** | **3.591 ± 1.197** | **5.960 ± 1.180** | **1.155 ± 0.706** | **4.420 ± 0.935** | **8.904 ± 1.214** | **2.386 ± 0.725** | **7.819 ± 1.212** |

Table 7: Comparison between two-stage and MOGA on News Dataset (mean ± standard deviation). Lower is better.

| Dataset | News-2 | | | News-4 | | | News-8 | | |
|---|---|---|---|---|---|---|---|---|---|
| Metric | $\sqrt{\hat{\epsilon}_{mPEHE}}$ | $\hat{\epsilon}_{mATE}$ | $\sqrt{AMSE}$ | $\sqrt{\hat{\epsilon}_{mPEHE}}$ | $\hat{\epsilon}_{mATE}$ | $\sqrt{AMSE}$ | $\sqrt{\hat{\epsilon}_{mPEHE}}$ | $\hat{\epsilon}_{mATE}$ | $\sqrt{AMSE}$ |
| Two-stage | 5.218 ± 1.079 | 0.741 ± 0.627 | 3.999 ± 0.758 | 6.126 ± 1.087 | 1.360 ± 0.747 | 5.968 ± 0.791 | 9.095 ± 0.841 | 2.788 ± 0.487 | 8.257 ± 0.687 |
| **MOGA** | **5.081 ± 1.693** | **0.449 ± 0.344** | **3.591 ± 1.197** | **5.960 ± 1.180** | **1.155 ± 0.706** | **4.420 ± 0.935** | **8.904 ± 1.214** | **2.386 ± 0.725** | **7.819 ± 1.212** |

### K.5 TWO-STAGE METHOD ON DISTANCE LEARNING

In this experiment, we modify the distance learning method in Section 3.3 to a two-stage approach. In the first stage, only the optimal transport matrices $\gamma_t$ are calculated. In the second stage, with $\gamma_t$ fixed, the matrices $\Theta_t$ are computed and then used to derive the mapping function parameterized by the matrix $\mathbf{P}$. Table 7 shows the results of the two-stage strategy and the original joint learning method. We observe that our proposed joint learning strategy achieves better performance compared with the two-stage strategy.

### K.6 PERFORMANCE OF DIFFERENT NUMBERS OF TREATMENTS

We provide the performance and running time results under different numbers of treatment $T$ in Table 8. Our method maintains competitive performance as the value of $T$ increases. Although the running time grows, our method still has a modest running time. If $T$ is extremely large, we can accelerate the distance learning stage by considering a subset of samples, and accelerate optimal transport by some fast algorithms (Gasteiger et al., 2021; Nguyen et al., 2022).

## L SENSITIVITY ANALYSIS

We take News-4 as an example to evaluate the effects of the hyper-parameters in our model. Figure 2 shows the results in terms of mPEHE with varying values of the hyperparameters $\lambda_h$, $\lambda_f$, $\lambda_y$, and $\rho$. From Figure 2(a), the performance decreases with a large $\lambda_h$, since a large $\lambda_h$ will induce a uniform transport plan $\gamma$, which cannot reflect different matching degrees based on the distances between samples. Figure 2(c) shows that a large $\lambda_y$ is helpful to achieve a better performance, which demonstrates the advantage of incorporating factual outcomes for distance learning. From Figures 2(b) and 2(d), we observe that our method performs stably in a wide range of values of $\lambda_f$ and $\rho$.

We also conduct ablation studies by setting the value of $\lambda_h$, $\lambda_f$ or $\lambda_y$ as 0. The results are also shown in Figure 2. We observe that the performance with $\lambda_h = 0$ and $\lambda_f = 0$ is worse compared with non-zero values, which verifies the effects of the entropic and Frobenius regularizations in Problem 11. $\lambda_y > 0$ achieves better performance compared with that of $\lambda_y = 0$, which demonstrates the effectiveness of introducing factual outcomes for distance learning.

Table 8: Results under different $T$ Settings (mean $\pm$ standard deviation). Lower is better.

| Setting | T=5 | | | | T=10 | | | | T=15 | | | |
|---|---|---|---|---|---|---|---|---|---|---|---|---|
| Metric | $\sqrt{\epsilon_{mPEHE}}$ | $\hat{\epsilon}_{mATE}$ | $\sqrt{AMSE}$ | Time (s) | $\sqrt{\epsilon_{mPEHE}}$ | $\hat{\epsilon}_{mATE}$ | $\sqrt{AMSE}$ | Time (s) | $\sqrt{\epsilon_{mPEHE}}$ | $\hat{\epsilon}_{mATE}$ | $\sqrt{AMSE}$ | Time (s) |
| MitNet | $1.3106 \pm 0.0407$ | $0.1529 \pm 0.0282$ | $1.1663 \pm 0.0269$ | 2471.059 | $1.4304 \pm 0.0161$ | $0.1225 \pm 0.0153$ | $1.1824 \pm 0.0084$ | 4578.611 | $1.5034 \pm 0.0152$ | $0.1161 \pm 0.0057$ | $1.1800 \pm 0.0095$ | 20736.247 |
| CFR | $1.9540 \pm 0.6629$ | $0.5958 \pm 0.3287$ | $2.2412 \pm 0.5553$ | 988.176 | $2.0670 \pm 0.6659$ | $0.5175 \pm 0.1409$ | $1.8434 \pm 0.4563$ | 3104.937 | $2.1712 \pm 0.4561$ | $0.3838 \pm 0.1842$ | $1.6091 \pm 0.3099$ | 7517.339 |
| PSM | $2.0315 \pm 0.2683$ | $1.0457 \pm 0.3283$ | $1.8500 \pm 0.1106$ | 3.240 | $1.9235 \pm 0.2388$ | $0.9493 \pm 0.3087$ | $1.7141 \pm 0.0523$ | 1.149 | $1.9329 \pm 0.2002$ | $1.0369 \pm 0.2332$ | $1.6883 \pm 0.0473$ | 21.015 |
| kNN | $1.4964 \pm 0.0680$ | $0.2546 \pm 0.0658$ | $1.1388 \pm 0.0471$ | 6.025 | $1.6412 \pm 0.0515$ | $0.2552 \pm 0.0557$ | $1.2603 \pm 0.0343$ | 14.950 | $1.6769 \pm 0.0283$ | $0.2451 \pm 0.0334$ | $1.2692 \pm 0.0160$ | 42.981 |
| GOM | $1.3903 \pm 0.0539$ | $0.4269 \pm 0.0957$ | $1.2028 \pm 0.0395$ | 1.266 | $1.4842 \pm 0.0353$ | $0.3070 \pm 0.0677$ | $1.2105 \pm 0.0250$ | 2.793 | $1.5404 \pm 0.0175$ | $0.2472 \pm 0.0245$ | $1.2001 \pm 0.0106$ | 6.410 |
| KOM | $1.6314 \pm 0.0591$ | $0.3546 \pm 0.0687$ | $1.3185 \pm 0.0393$ | 1.328 | $1.5474 \pm 0.0181$ | $0.1424 \pm 0.0367$ | $1.2460 \pm 0.0141$ | 2.912 | $1.5718 \pm 0.0164$ | $\mathbf{0.0831 \pm 0.0110}$ | $1.2196 \pm 0.0103$ | 2.964 |
| MOGA | $\mathbf{1.2962 \pm 0.0444}$ | $0.4606 \pm 0.0696$ | $1.1687 \pm 0.0287$ | 19.477 | $\mathbf{1.0885 \pm 0.0415}$ | $0.3320 \pm 0.0498$ | $1.1802 \pm 0.0112$ | 202.245 | $\mathbf{1.0356 \pm 0.0535}$ | $0.3009 \pm 0.0435$ | $1.1757 \pm 0.0092$ | 300.579 |

Table 9: Computation Time Comparison (in milliseconds).

| Method | Calculation Time (ms) |
|---|---|
| k-NN | 26.61 |
| OLS/LR-2 | 1528.69 |
| BART | 1806.28 |
| TARNet | 2094.19 |
| CFR | 2415.28 |
| GANITE | 20477.37 |
| PSM | 146.81 |
| PM | 5565.81 |
| CP | 1619.49 |
| MitNet | 29507.23 |
| MOGA | 4028.86 |

## M    VISUALIZATION OF OPTIMAL TRANSPORT PLAN

Figure 3 is the visualization of the optimal transport plan $\gamma$ in Eq. 11. Similar to simulation data generation in Section I, we generate data of $\boldsymbol{m} = [0.1, 0.2, 0.3]$ and $\boldsymbol{\Sigma}_{\text{rand}} \sim \mathcal{U}((0, [1, 1.5, 2]^\top)^{d \times d})$. We observe that $\gamma$ is meaningful and far away from a uniform coupling.

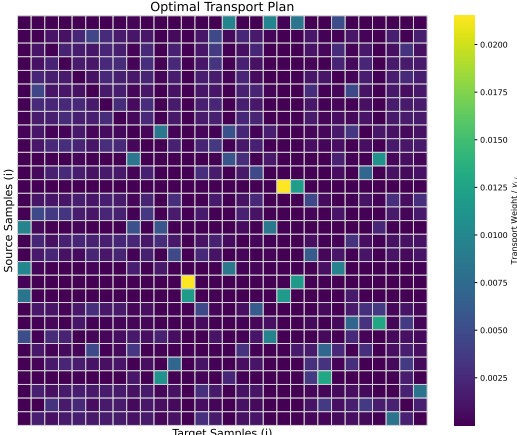

Figure 3: Visualization of the optimal transport plan. Lighter colors indicate larger weights, while darker colors represent weights closer to zero.

## N    RUNNING TIME RESULTS

We also provide the running times of representative methods in Table 9. We observe that the calculation time of our method is comparable to other methods.

