# OpenReview forum: "Matching without Group Barrier for Heterogeneous Treatment Effect Estimation"
_ICLR.cc/2026/Conference — ICLR 2026 Poster_

### Official Review · Reviewer_dnf7 · 2025-10-27

**Soundness:** 2
**Presentation:** 3
**Contribution:** 3
**Rating:** 4
**Confidence:** 5

**Summary:**

This paper investigate the limitation of existing matching approaches in causal inference, which select neighbors only from the target treatment group, leading to suboptimal counterfactual predictions due to data sparsity and distribution shifts. Hence, this paper proposes a novel matching framework: The core innovation is removing group barriers entirely, allowing neighbor selection from all samples regardless of treatment assignment. Then it develops a self optimal transport model and information propagation mechanism to realize the algorithm. The method is rigorously evaluated across both synthetic and semi-synthetic datasets.

**Strengths:**

1. The paper identifies a clear limitation in conventional matching methods-the data sparsity in the specific treatment group can induce the difficulty in finding the near neighborhood.

2. It provides a solid theoretical motivation by analyzing the estimation error in matching, which naturally leads to the formulation of the self optimal transport model.

3. The experimental design is thorough. A lot of typical baselines are incoporated into the experiments. This shows that the approach is effective.

**Weaknesses:**

1. This paper propose the algorithm based on the matching method. Although I believe the proposed method is superior to the vanilla matching method, it is still unclear why the matching-based method is superior to the other classes of methods, such as regression-based method (e.g. CFRNet and GANITE).

2. In the simulation experiments, only the statistics over all the setup $m=0.1, 0.4, 0.7, 1.0, 1.3$ is calculated. I suggest to reveal the results on each setup respectively. This is beneficial for understanding the behavior of the method under different scenarios.

**Questions:**

1. I am curious about the role of $Y \odot M$ in equation (14). I think $Y \odot M = Y$. Why does this term is incorporated?

2. In the current framework, the mapping function $\phi$ and the transport cost matrix are learned simultaneously. I am interested in exploring the effect of adopting a two-stage learning process where these two components are learned separately.

---

> ### Author Response · Authors · 2025-11-27
>
> We sincerely appreciate the valuable comments from the reviewers and provide point-by-point responses below. We will revise the manuscript accordingly based on the reviews and our responses.
>
> **Weakness 1:**
> Advantages compared with regression-based methods.
>
> **Answer:**
> Thank you for the valuable question.
> The strengths of our method rely on the following aspects:
>
> 1. Our method learns a distance metric that is directly guided by the factual outcomes, ensuring that the learned distance reflects meaningful relationships of potential outcomes of samples.
>
> 2. By leveraging the underlying manifold structure of the data, our approach can identify more appropriate and more stable matching counterparts.
>
> 3. Some regression-based methods seek to learn balanced representations, which could suffer from the over-balancing issue that informative information could be lost [a].
> Our method does not rely on balanced representation learning, alleviating the over-balancing issue.
>
> **Weakness 2:**
> The statistics on each setup.
>
> **Answer:**
> Thank you for the comments.
> We reported the mean and standard deviation results for all the setups of semi-synthetic and simulation data in Tables 1 and 2 on page 9.
>
> To ensure we address your concern accurately, we would be grateful if you could clarify what additional results are required to better understand the behavior of our method.
>
> **Question 1:**
> Explanation of Eq. (14) about $Y \odot M$.
>
> **Answer:**
> Thank you for the valuable comment.
> Yes, $Y \odot M = Y$. We explicitly write $Y \odot M$ to emphasize the masks $(1-M)$ and $M$.
> We will explain this and simplify $Y \odot M$ to $Y$ in the revision.
>
> **Question 2:**
> The results of a two-stage process to learn the mapping function $\phi$ and the transport plan.
>
> **Answer:**
> Thanks for your helpful advice.
> We try to modify DISTANCE LEARNING to a two-stage learning.
> Specifically,
>
> Stage 1: Only the optimal transport matrices $\gamma_t$ are calculated.
>
> Stage 2: With the $\gamma_t$ fixed, the matrices $\Theta_t$ are computed and then used to derive the mapping function parameterized by the matrix $P$.
>
> The following is the result of the two-stage strategy.
> We observe that our proposed joint learning strategy achieves better performance compared with the two-stage strategy.
>
> |      | News-2 (mpehe) | News-2 (mate) | News-2 (amse) | News-4 (mpehe) | News-4 (mate) | News-4 (amse) |
> | ------------- | -------------- | ------------- | ------------- | -------------- | ------------- | ------------- |
> | Two-stage | 5.218 ± 1.079  | 0.741 ± 0.627 | 3.999 ± 0.758 | 6.126 ± 1.087  | 1.360 ± 0.747 | 5.968 ± 0.791 |
> | MOGA      | 5.081 ± 1.693  | 0.449 ± 0.344 | 3.591 ± 1.197 | 5.960 ± 1.180  | 1.155 ± 0.706 | 4.420 ± 0.935 |
> |      |  **News-8 (mpehe)** | **News-8 (mate)** | **News-8 (amse)** |
> | Two-stage | 9.095 ± 0.841  | 2.788 ± 0.487 | 8.257 ± 0.687 |
> | MOGA      | 8.904 ± 1.214  | 2.386 ± 0.725 | 7.819 ± 1.212 |
>
>
> [a] Representation learning for treatment effect estimation from observational data, NeurIPS 2018.

---

### Official Review · Reviewer_MkPW · 2025-10-28

**Soundness:** 3
**Presentation:** 3
**Contribution:** 3
**Rating:** 6
**Confidence:** 3

**Summary:**

This paper proposes MOGA (Matching without Group Barrier), a novel method for estimating heterogeneous treatment effects (HTEs) under the potential outcomes framework.
The key innovation lies in establishing a theoretical and algorithmic connection between Sinkhorn-regularized optimal transport (OT) and counterfactual effect estimation.
Instead of restricting matching to samples within the same treatment group, MOGA finds neighbors across all groups, enabling more reliable counterfactual estimation even when group distributions differ.

The method further introduces a random walk–based outcome propagation procedure to iteratively diffuse factual outcomes over the learned sample graph, achieving global consistency of counterfactual prediction.
Additionally, MOGA learns a distance function via self-supervised optimal transport driven by factual outcomes, aligning the feature space with outcome similarity.

Experiments on semi-synthetic and simulated datasets show consistent improvements in PEHE, ATE, and AMSE metrics over a broad range of baselines.

**Strengths:**

- This paper formulates causal matching as a Sinkhorn-regularized optimal transport problem, and provides theoretical error bounds which explicitly motivate the design of the objective.
- The proposed framework integrates a novel random-walk mechanism for counterfactual propagation, combining matching, representation-learning and outcome-prediction into one unified pipeline. This global propagation over the sample manifold is inventive and extends beyond traditional nearest-neighbour or single‐group matching methods.

**Weaknesses:**

- The empirical evaluation is restricted to semi-synthetic and simulation datasets. While these are common in causal inference research, including at least one real-world biased observational dataset would significantly strengthen the practical claims of the method, ensuring it generalises beyond controlled or synthetic settings.
- The experimental comparisons omit recent OT-based causal estimators.
- The current formulation appears to assume relatively homogeneous treatment-group distributions and does not explicitly model or correct for internal biases within treatment groups or highly unequal propensity distributions.

**Questions:**

- How would the method perform on real-world biased observational datasets, or what challenges prevent such evaluation?
- How does the approach address intra-group bias or unequal propensity distributions that may exist within treatment groups?
- The Sinkhorn regularization introduces an entropy term that can make the transport plan overly smooth, potentially blurring the matching structure. How sensitive is the method to the entropy coefficient $\lambda_h$? Have the authors considered strategies to prevent the transport plan from collapsing into a nearly uniform coupling?

---

> ### Author Response · Authors · 2025-11-27
>
> We sincerely appreciate the valuable comments from the reviewers and provide point-by-point responses below. We will revise the manuscript accordingly based on the reviews and our responses.
>
> **Weakness 1 and Question 1:**
> Results on real-world observational data.
>
> **Answer:**
> Thanks for your valuable comments.
> We present the performance of MOGA alongside other baseline approaches on the Lalonde dataset [a], which is a representative real-world dataset.
> Our proposed method still obtains promising results.
>
> |        | mae of Lalonde       |
> |--------|----------------------|
> | MitNet | 393.9513 ± 87.6213   |
> | CFR    | 495.3089 ± 59.5542  |
> | PSM    | 728.7922 ± 500.1740 |
> | kNN    | 275.2712 ± 148.3843 |
> | GOM    | 434.4803 ± 231.6803 |
> | KOM    | 438.3496 ± 255.4207 |
> | CEM    | 796.7605 ± 610.0612 |
> | MOGA   | 246.1526 ± 86.9842  |
>
>
> **Weakness 2:**
> Comparison with recent OT-based methods.
>
> **Answer:**
> Thanks for your insightful suggestions.
> We compare MOGA with EsCFR [b] and CE-RCFR [c] which are based on OT as follows.
> Our method obtains promising performance compared with others.
>
> |    | News-2 (pehe) | News-2 (ate) | News-2 (amse) | News-4 (mpehe) | News-4 (mate) | News-4 (amse) |
> | ------- | -------------- | ------------- | ------------- | -------------- | ------------- | ------------- |
> | CE-RCFR | 9.316 ± 2.397  | 1.613 ± 1.561 | 6.972 ± 1.836 | 9.458 ± 1.828  | 2.791 ± 1.179 | 8.017 ± 1.736 |
> | EsCFR   | 6.711 ± 1.314  | 1.024 ± 1.431 | 4.224 ± 1.728 | 6.581 ± 1.974  | 2.121 ± 1.429 | 4.219 ± 1.652 |
> | MOGA    | 5.081 ± 1.693  | 0.449 ± 0.344 | 3.591 ± 1.197 | 5.960 ± 1.180  | 1.155 ± 0.706 | 4.420 ± 0.935 |
> |    | **News-8 (mpehe)** |  **News-8 (mate)** | **News-8 (amse)** |
> | CE-RCFR | 10.545 ± 1.429 | 4.057 ± 1.172 | 10.506 ± 1.855 |
> | EsCFR   | 9.611 ± 1.289  | 3.214 ± 1.642 | 7.646 ± 1.623  |
> | MOGA    | 8.904 ± 1.214  | 2.386 ± 0.725 | 7.819 ± 1.212  |
>
>
> **Weakness 3 and Question 2:**
> Regarding internal bias and unequal propensity distributions.
>
> **Answer:**
> Thank you for the valuable comment.
>
> 1. For the internal bias, our optimal transport model implicitly mitigates this bias.
> First, we learn a distance metric supervised by factual outcomes, which captures internal heterogeneity and enhances the property that close samples with a small learned distance have similar outcomes.
> Second, our optimal transport method assigns equal marginal mass to each sample, preventing large subgroups from dominating small subgroups, which ensures that internal subgroups maintain influence in the matching process.
>
> 2. For unequal propensity distributions, this may pose a challenge that matched samples are not close enough, resulting in outcome estimation error.
> Our method is designed to alleviate this issue by expanding the matching pool, which is achieved by considering all groups rather than only the target group as candidate samples.
> The results in Table 2 support this: $m$ controls the distribution means, and the more spread out $m$ is, the less the group distributions overlap and the worse all methods perform.
> Our method still remains competitive, which demonstrates the robustness of our method.
>
> **Question 3:**
> Sensitivity to the entropy coefficient $\lambda_h$ and how to prevent the transport plan from collapsing into a nearly uniform coupling.
>
> **Answer:**
> Thank you for the insightful suggestion.
> We have analyzed the parameter sensitivity of $\lambda_h$ of the entropic parameter in Section L and Figure 2 on pages 21 and 22.
> Our method is insensitive to this $\lambda_h$ and maintains a good performance in a wide range.
>
> When the entropic parameter $\lambda_h$ is extremely large, the optimal transport plan could collapse to a nearly uniform coupling.
> Usually, when $\lambda_h$ is in a reasonable range, the optimal transport plan will not collapse.
> In https://anonymous.4open.science/r/iclr-submit-MOGA-B8A1/ot_plan_visualization.pdf,
> we also visualize the learned optimal transport matrix $\gamma$.
> We generate the data of $\boldsymbol{m}=[0.1,0.2,0.3]$ and $\Sigma_{\text{rand}} \sim \mathcal{U}((0,[1,1.5,2]^\top)^{d \times d})$.
> We observe that $\gamma$ is meaningful and far away from a uniform coupling.
>
> [a] Evaluating the econometric evaluations of training programs with experimental data, The American Economic Review.
>
> [b] Optimal transport for treatment effect estimation, NeurIPS 2023.
>
> [c] CE-RCFR: Robust counterfactual regression for consensus-enabled treatment effect estimation, SIGKDD 2024.

---

> > ### Comment · Reviewer_MkPW · 2025-11-28
> >
> > Thanks authors for the detailed response. I am satisfied with the addition of results on the Lalonde dataset and the comparison with recent OT-based baselines (EsCFR, CE-RCFR).
> >
> > Regarding Weakness 3/Question 2 (Intra-group Bias), while I accept that the equal marginal mass and outcome-supervised distance implicitly address bias, I still recommend the authors consider integrating an explicit mechanism into the distance learning objective $\mathcal{L}_H$. This would enhance the theoretical robustness by explicitly forcing covariate alignment.
> >
> > Overall, The MOGA framework is a clever combination of OT and outcome propagation, and I will hold my positive score.

---

### Official Review · Reviewer_7WJ5 · 2025-10-30

**Soundness:** 2
**Presentation:** 3
**Contribution:** 2
**Rating:** 4
**Confidence:** 4

**Summary:**

This paper addresses a fundamental challenge in heterogeneous treatment effect estimation—the unobservability of counterfactual outcomes. Targeting the issue of inadequate matching quality caused by within-group constraints in traditional matching methods, it proposes a Matching method named as Matching withOut Group bArrier (MOGA). MOGA introduces a matching framework that transcends inter-group barriers. While conventional matching only seeks nearest neighbors within the target treatment group, MOGA innovatively allows matching across all treatment groups. By optimizing matching weights through a self-optimal transport model, it expands the effective matching pool and mitigates the problem of excessive matching distances caused by data distribution discrepancies or insufficient samples. Unlike traditional matching that relies solely on covariates, MOGA incorporates factual outcomes into distance learning, ensuring that the matching distance aligns with outcome relevance. Furthermore, it enhances the smoothness of counterfactual predictions by disseminating outcome information via random walks and leveraging the manifold structure of the data, thereby balancing covariate similarity with consistency in outcome patterns.

**Strengths:**

In terms of originality, this paper addresses the limitations of traditional within-group matching methods by proposing the MOGA framework, which enables cross-group matching. It integrates an optimal transport model with an outcome propagation mechanism, thereby forming a methodological adjustment to the existing matching paradigm. Regarding quality, the method is designed based on theoretical foundations such as the error bound of optimal transport optimization. Experiments conducted on semi-synthetic and simulated datasets, benchmarking against baseline methods like k-NN and PSM, demonstrate its relative advantages in metrics such as PEHE and ATE. The contribution of individual modules is further validated through ablation studies. In terms of clarity, the paper exhibits a logically coherent structure, with comprehensive explanations of technical details (e.g., the optimal transport formulation) and well-defined key concepts (e.g., cross-group matching), ensuring strong comprehensibility. As for significance, the study addresses the inadequacy of within-group matching in estimating heterogeneous treatment effects from observational data by offering a solution that broadens the matching pool. It enhances counterfactual prediction accuracy in multi-treatment scenarios, thereby holding referential value for methodological applications in the field of causal inference.

**Weaknesses:**

1. Insufficient Verification of Theoretical Assumptions.
The core of MOGA relies on the "self-optimal transport model" and the "outcome propagation mechanism." However, the paper does not adequately justify the rationality and applicability boundaries of its key assumptions. Optimal transport (OT) models require the transport cost (i.e., the distance between samples) to satisfy mathematical properties such as convexity and lower semicontinuity, while also assuming a certain level of smoothness in the data distribution. The paper does not explicitly clarify whether real-world data meet these conditions, nor does it verify whether the OT model may fail (e.g., by concentrating matching weights on a few samples) when the data exhibit high-dimensional sparsity, non-convex distributions, or outliers, thereby compromising the stability of counterfactual predictions.
2. Practical application is questionable.
The computational complexity of optimal transport models is typically . Although the paper mentions the use of approximation algorithms for optimization, it does not report specific computational time or memory usage. Furthermore, in multi-treatment scenarios, the dimensionality of the transport matrix expands from  to , leading to an exponential increase in complexity. The paper fails to provide optimization strategies or experimental validation for such scenarios, raising doubts about the scalability of the method in practical applications.
3. Insufficient Experimental Validation.
The paper only compares MOGA with k-NN, PSM, and TARNet, without including more advanced methods in the current HTE field (e.g., DragonNet, R-learner, or the Meta-learner series) or improved matching-based methods (e.g., Covariate Balancing Propensity Score). This makes it difficult to demonstrate whether the "significant advantages" of MOGA stem from methodological innovation or baseline selection bias. Additionally, the experiments primarily rely on semi-synthetic and simulated data. Semi-synthetic data, often generated based on known models, may fail to capture real-world challenges such as high-dimensional nonlinear confounding and sample selection bias.

**Questions:**

1. The optimal transport models relied upon by MOGA typically assume that data distributions satisfy smoothness or convexity assumptions. If the data are high-dimensional, sparse, non-convex, or contain outliers, might the transport plans introduce biases that affect the stability of counterfactual predictions?

2. The experiments are primarily based on semi-synthetic and simulated data. Has MOGA been validated on real observational data? If so, do metrics such as PEHE and ATE align with those obtained from synthetic data? If not, how can the method's generalization capability be ensured in real-world scenarios with unobservable counterfactuals?

3. The paper claims MOGA exhibits enhanced stability in multi-treatment settings; however, it does not specify the exact number of treatments evaluated (e.g., K=3, 5, 10) nor the associated computational complexity as K increases. Could you provide performance comparisons across different K values and elaborate on scalability optimization strategies for scenarios where K > 10?

4. Could you provide comparative results benchmarking MOGA against more advanced methodologies, such as DragonNet, R-learner, and Meta-learners?

5. MOGA expands the matching pool through cross-group matching, which may implicitly impose stronger exchangeability requirements. Has sensitivity analysis  been conducted to quantify its robustness to unobserved confounding?

---

> ### Author Response · Authors · 2025-11-27
>
> We sincerely appreciate the valuable comments from the reviewers and provide point-by-point responses below. We will revise the manuscript accordingly based on the reviews and our responses.
>
> **Weakness 1 and Question 1:**
> Discussion on complex data situations.
>
> **Answer:**
> Thank you for the valuable comment.
>
> 1. Regarding high-dimensional or sparse data:
> In Section 3.3, we learn a distance function with the guidance of the factual outcomes, and implement the distance function in a learnable projected subspace in Eq. (19).
> As a result, the optimal transport is performed in a supervised subspace, which can alleviate the issue of high-dimensional or sparse data.
>
> 2. Regarding data distribution:
> One of the advantages of optimal transport is that it can be performed without knowing the underlying distribution.
> Optimal transport can work even on discrete distributions represented by empirical samples.
>
> 3. Regarding outliers:
> The optimal transport plan reflects the matching degrees used for counterfactual outcome estimation.
> Optimal transport will adaptively assign larger values $\gamma_{ij}$ between close sample pairs, and a pair far away from each other will receive a quite small $\gamma_{ij}$.
> For outliers that are far away from most samples, optimal transport will assign small weights for them.
> As a result, the estimated outcomes are basically determined by close samples with large weights, and the outliers will make a limited contribution in counterfactual outcome estimation.
>
>
> **Weakness 2**
> Computational complexity and memory usage.
>
> **Answer:**
> Thank you for your insightful comments.
> We discuss the computational complexity and memory as follows.
>
> **1. Computational Complexity:**
> Although our method involves solving optimal transport problems,
> we do not need to train a neural network,
> and distance learning enjoys a closed-form solution.
> Therefore, the computational complexity of our method is acceptable.
>
> For Algorithm 1,
> let $N$ and $n_t$ be the numbers of all the samples and the samples in the treatment group $t$,
> $d$ and $d'$ be the numbers of the features before and after projection.
> For distance learning of the group $t$, the complexity of Step 3 is $O(n_t^2d + n_td^2 + d^3)$,
> the complexity of Step 4 is $O(n_t^2d')$.
> For optimal transport matching, the complexity of Steps 6 and 7 is $O(Ndd' + N^2d')$.
> For counterfactual prediction, the complexity of Step 9 is $O(N^2T)$,
> where $T$ is the number of the different treatment values.
>
> Furthermore, the following table reports the execution times of the conducted methods on a synthetic dataset.
> Our method achieves modest time efficiency.
>
> |          | Calculation time |
> |----------|------------------|
> | k-NN     | 26.61 ms         |
> | OLS/LR-2 | 1528.69 ms       |
> | BART     | 1806.28 ms       |
> | TARNet   | 2094.19 ms       |
> | CFR      | 2415.28 ms       |
> | GANITE   | 20477.37 ms      |
> | PSM      | 146.81 ms        |
> | PM       | 5565.81 ms       |
> | CP       | 1619.49 ms       |
> | MitNet   | 29507.23 ms      |
> | MOGA     | 4028.86 ms       |
>
>
> **2. Space Complexity:**
> The overall space complexity of the algorithm is dominated by the storage of the $N \times N$ transition probability matrix $W$,
> as highlighted in the following two steps:
>
> (1) Optimal Transport Plan Calculation.
> First, during the calculation of the Optimal Transport (OT) Plan to build the transition probability matrix $W$, the primary memory requirement is for simultaneously storing the input data $X$: $O(Nd)$, where $d$ is the feature dimension, and the transition probability matrix $W$: $O(N^2)$.
> This step's total space complexity is $\text{Space Complexity} = O(N^2 + N d)$.
> %As the sample size $N$ is typically much larger than the feature dimension $d'$ in this context, the complexity is determined by the quadratic term, simplifying to $O(N^2)$.
>
> (2) Label Propagation Subsequently.
> During label propagation, the essential operation involves matrix multiplication, $S Y$, where $S \in \mathbb{R}^{N \times N}$ is the affinity matrix, and $Y$ is the $N \times T$ label matrix.
> The total space required during this multiplication operation must accommodate the largest input matrix $W$ ($N \times N$) along with the inputs $Y$ ($N \times T$) and the output $W \cdot Y$ ($N \times T$). Therefore, the peak space complexity for this stage is: $\text{Space}_{\text{Total}} = O(N^2 + N \times T + N \times T) = O(N^2)$.
>
> (3) In summary, the algorithm's dominant space complexity is consistently $O(N^2 + Nd)$.

---

> > ### Author Response · Authors · 2025-11-27
> >
> > **Weakness 3 and Question 4:** Comparison with more advanced methods.
> >
> > **Answer:**
> > Thank you for your valuable suggestion.
> > We compare MOGA with DragonNet, Meta-learner, CBPS (Covariate Balancing Propensity Score) and MALTS [b] on the News dataset as follows.
> > We observe that our method achieves promising performance
> > | Method    | News-2 (pehe) | News-2 (ate) | News-2 (amse) | News-4 (mpehe) | News-4 (mate) | News-4 (amse)  |
> > | --------- | -------------- | ------------- | ------------- | -------------- | ------------- | -------------- |
> > | DragonNet | 9.408 ± 2.324  | 1.990 ± 1.567 | 6.969 ± 1.838 | 10.359 ± 4.153 | 3.787 ± 3.960 | 8.015 ± 2.611  |
> > | R-learner | 9.336 ± 2.367  | 1.710 ± 1.521 | 6.902 ± 2.276 | 9.646 ± 2.048  | 3.118 ± 1.654 | 8.479 ± 2.212  |
> > | S-learner | 8.646 ± 2.374  | 1.533 ± 1.468 | 6.386 ± 1.743 | 8.993 ± 1.846  | 2.591 ± 1.282 | 7.845 ± 1.747  |
> > | T-learner | 8.381 ± 2.246  | 1.581 ± 1.512 | 6.258 ± 1.723 | 8.800 ± 1.846  | 2.658 ± 1.273 | 7.774 ± 1.747  |
> > | X-learner | 8.577 ± 2.270  | 1.574 ± 1.510 | 6.329 ± 1.732 | 8.825 ± 1.842  | 2.661 ± 1.275 | 7.789 ± 1.748  |
> > | CBPS      | 10.646 ± 2.847 | 2.872 ± 2.439 | 7.528 ± 2.013 | 13.214 ± 2.679 | 2.891 ± 1.763 | 10.929 ± 2.679 |
> > | MALTS     | 10.692 ± 2.663 | 1.553 ± 1.486 | 7.911 ± 2.025 | 10.382 ± 1.996 | 2.658 ± 1.273 | 8.713 ± 1.833  |
> > | MOGA      | 5.081 ± 1.693  | 0.449 ± 0.344 | 3.591 ± 1.197 | 5.960 ± 1.180  | 1.155 ± 0.706 | 4.420 ± 0.935  |
> > | **Method**    | **News-8 (mpehe)** | **News-8 (mate)** | **News-8 (amse)**  |
> > | DragonNet | 16.904 ± 5.637 | 8.503 ± 7.913 | 18.367 ± 6.896 |
> > | R-learner | 10.470 ± 1.282 | 3.867 ± 0.908 | 11.035 ± 1.743 |
> > | S-learner | 10.320 ± 1.313 | 3.900 ± 0.951 | 10.577 ± 1.670 |
> > | T-learner | 10.284 ± 1.297 | 3.914 ± 0.927 | 10.582 ± 1.645 |
> > | X-learner | 10.250 ± 1.293 | 3.922 ± 0.925 | 10.570 ± 1.643 |
> > | CBPS      | 15.817 ± 1.987 | 3.437 ± 0.974 | 14.845 ± 2.385 |
> > | MALTS     | 13.123 ± 1.890 | 3.278 ± 1.439 | 10.224 ± 2.435 |
> > | MOGA      | 8.904 ± 1.214  | 2.386 ± 0.725 | 7.819 ± 1.212  |
> >
> >
> >
> > **Question 2:**
> > Results of real-world data and more advanced methods.
> >
> > **Answer:**
> > Thanks for your valuable comments.
> > We present the performance of MOGA alongside other baseline approaches on the Lalonde dataset [a], which is a representative real-world dataset.
> > Our proposed method still obtains promising results.
> >
> > |        | mae of Lalonde      |
> > |--------|---------------------|
> > | MitNet | 393.9513 ± 87.6213  |
> > | CFR    | 495.3089 ± 59.5542  |
> > | PSM    | 728.7922 ± 500.1740 |
> > | kNN    | 275.2712 ± 148.3843 |
> > | GOM    | 434.4803 ± 231.6803 |
> > | KOM    | 438.3496 ± 255.4207 |
> > | CEM    | 796.7605 ± 610.0612 |
> > | MOGA   | 246.1526 ± 86.9842  |

---

> ### Author Response · Authors · 2025-11-27
>
> **Question 3:**
> Results of different numbers of treatments $K$.
>
> **Answer:**
> Thank you for the valuable suggestion.
> We provide the performance and running time results under different $K$ as follows.
> Our method maintains competitive performance as the value of $K$ increases.
> Although the running time grows, our method still has a modest running time.
>
> | K=5   | mpehe           | mate            | amse            | time       |
> | ------ | --------------- | --------------- | --------------- | ---------- |
> | MitNet | 1.3106 ± 0.0407 | 0.1529 ± 0.0282 | 1.1663 ± 0.0269 | 2471.059 s |
> | CFR    | 1.9540 ± 0.6629 | 0.5958 ± 0.3287 | 2.2412 ± 0.5553 | 988.176 s  |
> | PSM    | 2.0315 ± 0.2683 | 1.0457 ± 0.3283 | 1.8500 ± 0.1106 | 3.240 s    |
> | kNN    | 1.4964 ± 0.0680 | 0.2546 ± 0.0658 | 1.1388 ± 0.0471 | 6.025 s    |
> | GOM    | 1.3903 ± 0.0539 | 0.4269 ± 0.0957 | 1.2028 ± 0.0395 | 1.266 s    |
> | KOM    | 1.6314 ± 0.0591 | 0.3546 ± 0.0687 | 1.3185 ± 0.0393 | 1.328 s    |
> | MOGA   | 1.2962 ± 0.0444 | 0.4606 ± 0.0696 | 1.1687 ± 0.0287 | 19.477 s   |
> | **K=10**  | **mpehe**           | **mate**            | **amse**            | **time**        |
> | MitNet | 1.4304 ± 0.0161 | 0.1225 ± 0.0153 | 1.1824 ± 0.0084 | 4578.611 s |
> | CFR    | 2.0670 ± 0.6659 | 0.5175 ± 0.1409 | 1.8434 ± 0.4563 | 3104.937 s |
> | PSM    | 1.9235 ± 0.2388 | 0.9493 ± 0.3087 | 1.7141 ± 0.0523 | 1.149 s    |
> | kNN    | 1.6412 ± 0.0515 | 0.2552 ± 0.0557 | 1.2603 ± 0.0343 | 14.950 s   |
> | GOM    | 1.4842 ± 0.0353 | 0.3070 ± 0.0677 | 1.2105 ± 0.0250 | 2.793 s    |
> | KOM    | 1.5474 ± 0.0181 | 0.1424 ± 0.0367 | 1.2460 ± 0.0141 | 2.912 s    |
> | MOGA   | 1.0885 ± 0.0415 | 0.3320 ± 0.0498 | 1.1802 ± 0.0112 | 202.245 s  |
> | **K=15**   | **mpehe**           | **mate**            | **amse**            | **time**        |
> | MitNet | 1.5034 ± 0.0152 | 0.1161 ± 0.0057 | 1.1800 ± 0.0095 | 20736.247 s |
> | CFR    | 2.1712 ± 0.4561 | 0.3838 ± 0.1842 | 1.6091 ± 0.3099 | 7517.339 s  |
> | PSM    | 1.9329 ± 0.2002 | 1.0369 ± 0.2332 | 1.6883 ± 0.0473 | 21.015 s    |
> | kNN    | 1.6769 ± 0.0283 | 0.2451 ± 0.0334 | 1.2692 ± 0.0160 | 42.981 s    |
> | GOM    | 1.5404 ± 0.0175 | 0.2472 ± 0.0245 | 1.2001 ± 0.0106 | 6.410 s     |
> | KOM    | 1.5718 ± 0.0164 | 0.0831 ± 0.0110 | 1.2196 ± 0.0103 | 2.964 s     |
> | MOGA   | 1.0356 ± 0.0535 | 0.3009 ± 0.0435 | 1.1757 ± 0.0092 | 300.579 s   |
>
>
>
> The distance learning in Section 3.3 is performed within each group.
> Therefore, the computational complexity of distance learning is linear with respect to the number of treatments $K$.
> For optimal transport matching in Problem (11), the complexity mainly depends on the number of all the samples $N$.
> For counterfactual prediction in Section 3.2, the complexity is also linear with respect to $K$.
>
> If $K$ is extremely large, we can accelerate the DISTANCE LEARNING stage by considering a subset of samples,
> and accelerate optimal transport by some fast algorithms [e][f].
>
> **Question 5:**
> The exchangeability requirement and the issue of unobserved confounding.
>
> **Answer:**
> Thank you for your insightful comment.
>
> 1. Our proposed method still relies on the standard conditional exchangeability assumption, i.e.,
> $Y_t(x_i)\perp t_i | x_i$, which is standard and implies the usual conditional exchangeability among groups.
> This assumption is sufficient for effect identification, since it allows the observed outcomes within covariate strata to serve as unbiased estimators of corresponding potential outcomes.
> No stronger assumptions are required for our method, and under this assumption, valid causal comparisons across treatment groups can be obtained.
>
> 2. Unobserved confounding poses difficulties for all the causal inference methods relying on standard assumptions.
> In this study, we still consider the standard assumption, including the unconfoundedness assumption.
> If the unconfoundedness assumption is violated, additional assumptions are required in existing studies, such as the presence of instrumental variables [c], or auxiliary random controlled trial data [d].
> Nevertheless, it is beyond the scope of our work, and we will clarify the needed assumptions more explicitly in our revised version.
>
> We will add the above discussion in the revision.
>
> [a] Evaluating the econometric evaluations of training programs with experimental data. The American Economic Review.
>
> [b] Malts: Matching after learning to stretch, JMLR 2022.
>
> [c] Estimating individualized causal effect with confounded instruments, SIGKDD 2022.
>
> [d] Removing hidden confounding by experimental grounding, NeurIPS 2018.
>
> [e] Scalable Optimal Transport in High Dimensions for Graph Distances, Embedding Alignment, and More. ICML 2021.
>
> [f] Improving Mini-batch Optimal Transport via Partial Transportation, ICML 2022.

---

### Official Review · Reviewer_VXGC · 2025-10-31

**Soundness:** 3
**Presentation:** 3
**Contribution:** 3
**Rating:** 6
**Confidence:** 3

**Summary:**

The paper introduces a novel approach for estimating heterogeneous treatment effects (HTE) using matching without group barriers. In traditional matching methods, neighbors for a treatment group are selected solely from within that same group. However, the authors argue that such methods are limited by distribution discrepancies and scarce samples in some treatment groups. To address this, the paper proposes a method that removes the barriers between treatment groups, selecting nearest neighbors from all available samples, improving counterfactual predictions. This method is modeled as a self-optimal transport problem, where the transport cost is measured using the distance between samples and factual outcomes. This new method, called Matching withOut Group Barrier (MOGA), is demonstrated to improve prediction accuracy, particularly in both binary and multiple treatment settings, with extensive experiments on synthetic and semi-synthetic datasets.

**Strengths:**

1.	Innovative Approach: The method removes the group barrier in matching, which is a common issue in traditional causal inference methods, allowing for better matching of similar samples from different treatment groups.

2.	Theoretical Foundation: The method is supported by a robust theoretical analysis, providing error bounds and drawing a connection with optimal transport.

3.	Comprehensive Experiments: The paper presents experiments on a variety of datasets, including semi-synthetic data and real-world datasets like TCGA, comparing the proposed method against several traditional methods (e.g., k-NN, PSM, BART). The experimental results demonstrate that MOGA outperforms traditional matching methods, as well as other advanced models like BART and TARNet, in terms of precision in estimating heterogeneous treatment effects.

**Weaknesses:**

1.	Complexity: The method involves solving optimal transport problems and learning transport plans, which may be computationally intensive, especially for large datasets. The algorithm requires significant computational resources (e.g., a high-performance GPU) .

2.	Dependence on Hyperparameters: The model's performance is sensitive to hyperparameters like λh, λf, λy, and ρ. While the authors provide sensitivity analysis, the method might require careful tuning depending on the dataset .

3.	Assumptions: A key assumption of the proposed method is that samples from different treatment groups can be meaningfully matched, which may not hold in real-world scenarios. In the presence of unobserved confounding, where latent variables affect both treatment assignment and outcomes, the matching may fail to balance treatment groups, leading to biased estimates. Moreover, if treatment effects vary significantly across subgroups, the assumption of similar effects for matched samples breaks down, as heterogeneous treatment effects may not be captured accurately. Finally, substantial distributional shifts between treatment groups can further hinder meaningful matching, particularly when groups are non-exchangeable due to underlying differences. Thus, the method's effectiveness is highly contingent on these assumptions, which may not always hold in practical settings.

**Questions:**

Refer to Weaknesses above.

---

> ### Author Response · Authors · 2025-11-27
>
> We sincerely appreciate the valuable comments from the reviewers and provide point-by-point responses below. We will revise the manuscript accordingly based on the reviews and our responses.
>
> **Weakness 1:**
> Computational complexity.
>
> **Answer:**
> Thank you for the insightful comment.
> Although our method involves solving optimal transport problems,
> we do not need to train a neural network,
> and distance learning enjoys a closed-form solution.
> Therefore, the computational complexity of our method is acceptable.
> We report the running time results as follows.
> We observe that the calculation time of our method is comparable to other methods.
>
> |          | Calculation time |
> |----------|------------------|
> | k-NN     | 26.61 ms         |
> | OLS/LR-2 | 1528.69 ms       |
> | BART     | 1806.28 ms       |
> | TARNet   | 2094.19 ms       |
> | CFR      | 2415.28 ms       |
> | GANITE   | 20477.37 ms      |
> | PSM      | 146.81 ms        |
> | PM       | 5565.81 ms       |
> | CP       | 1619.49 ms       |
> | MitNet   | 29507.23 ms      |
> | MOGA     | 4028.86 ms       |
>
>
> **Weakness 2:**
> Dependence on Hyperparameters.
>
> **Answer:**
> Thank you for the valuable comment.
> According to the results of the sensitivity analysis in Section L and Figure 2,
> our method can achieve promising performance in a wide range of hyperparameters.
> Therefore, it is not a significant challenge to tune the hyperparameters.
>
>
> **Weakness 3:**
> Discussion on the assumptions.
>
> **Answer:**
> We thank the reviewer for raising this point. We discuss the adopted assumptions in the following.
>
> 1. Indeed, it is challenging to find meaningfully matched samples for matching-based methods.
> However, compared with existing matching methods, our method enlarges the candidate pool, searching the possibly matched samples not only within the targeted treatment group but also across all groups. This effectively strengthens the matching strategies and substantially alleviates this issue.
>
>
> 2. Regarding the unobserved confounding, we agree that unobserved confounding poses difficulties to all the causal inference methods relying on standard identifiability assumptions.
> In this study, we still consider the standard assumption, including the unconfoundedness assumption.
> If the unconfoundedness assumption is violated, additional assumptions are required in existing studies, such as the presence of instrumental variables [a], or auxiliary random controlled trial data [b].
> Nevertheless, it is beyond the scope of our work, and we will clarify the needed assumptions more explicitly in our revised version.
>
> 3. For the treatment effect varying across subgroups, our proposed distance learning method can alleviate this issue. In Section 3.3, we propose a method to learn an effective distance supervised by the factual outcomes. Such a method enhances the property that similar samples with a small learned distance have similar outcomes.
> Experimental results in Section K and Table 3 on Pages 20 and 21 support this. We observe that the learned distance performs better compared with other distances without outcome supervision.
>
> 4. We agree with the reviewer that the substantial distribution shift across groups poses a challenge for meaningful matching.
> However, our method is specifically designed to alleviate such a problem by expanding the candidate matching pool from one targeted group to all groups, increasing the possibility of finding closer units.
> The results in Table 2 support this: $m$ controls the distribution means, and the more spread out $m$ is, the less the group distributions overlap and the worse the performance of all the methods.
> Nevertheless, our method still remains competitive, which demonstrates the robustness of our method.
>
>
> [a] Estimating individualized causal effect with confounded instruments, SIGKDD 2022.
>
> [b] Removing hidden confounding by experimental grounding, NeurIPS 2018.

---

### Meta-Review · Area_Chair_Ys93 · 2026-01-07

**Summary:**

Limited evaluation realism and generalization: The initial submission relied mainly on synthetic and semi-synthetic data, which raised concerns about whether the method would generalize to real-world observational settings.

Insufficient baseline comparisons: Reviewers noted that comparisons with several strong and widely used HTE methods and advanced matching approaches were missing. It is difficult to clearly assess the practical advantage of the proposed method.

Computational cost and scalability: The use of optimal transport raised concerns about computational and memory requirements, especially in multi-treatment settings where the size of the transport plan grows with the number of treatments.

Strength and clarity of assumptions: Reviewers questioned whether cross-group matching is reliable under distribution shifts, unequal propensity distributions, or unobserved confounding, and asked for clearer discussion of the required assumptions and their limitations.

Stability and sensitivity of the OT formulation: Potential instability due to entropy regularization and sensitivity to hyperparameters were highlighted, including the risk of overly smooth or nearly uniform transport plans.

Clarity and completeness of analysis: Reviewers pointed out unclear technical details and requested more detailed analyses, such as results broken down by simulation settings and clearer explanations of specific equations and design choices.

**Reviewer Concerns:**

Evaluation realism and generalizability: The rebuttal addressed this concern by adding experiments on a real-world observational dataset, which directly responds to multiple reviewers’ requests. While this strengthens the empirical evidence, evaluation is still limited to a small number of real-world datasets.

Baseline comparisons: The rebuttal substantially addressed this concern by adding comparisons with strong and widely used HTE methods as well as recent OT-based approaches. This resolves the reviewers’ concerns.

Computational cost and scalability: The rebuttal clarified computational and memory complexity, reported runtime results, and added experiments with increasing numbers of treatment groups. These additions address the feasibility concerns raised by reviewers, although scalability to very large datasets or very large numbers of treatments remains a practical limitation.

Assumptions and robustness: The rebuttal clarified that the method relies on standard conditional exchangeability and unconfoundedness assumptions and discussed their scope and limitations. This addressed the lack of clarity noted by reviewers.

OT stability and hyperparameter sensitivity: The rebuttal addressed concerns about entropy regularization and hyperparameter sensitivity by adding sensitivity analyses and visualizations of the learned transport plans, which alleviated concerns about degenerate or overly smooth couplings.

Clarity and completeness of analysis: The rebuttal improved clarity by explaining specific equations, adding analyses such as joint versus two-stage learning, and expanding experimental results. Some requests for more fine-grained breakdowns of simulation results were partially addressed.

**Reviewer Scores:**

Overall, the rebuttal addressed most concrete concerns raised in the reviews, particularly by adding real-world experiments, stronger baseline comparisons, and clearer analyses of complexity, assumptions, and stability.
Reviewers who were initially positive would likely maintain their scores and possibly gain increased confidence in the soundness and empirical support of the method. Reviewers with borderline evaluations would likely shift slightly toward a more favorable assessment, as many of their practical and empirical concerns were directly addressed.

---

### Decision · Program_Chairs · 2026-01-26

Accept (Poster)